# FANCJ DNA helicase is recruited to the replisome by AND-1 to ensure genome stability

Ana Boavida [1,2], Luisa MR Napolitano[3], Diana Santos [1,2], Giuseppe Cortone[1], Nanda K Jegadesan [4], Silvia Onesti [3], Dana Branzei [4,5] & Francesca M Pisani [1✉]

## Abstract

FANCJ, a DNA helicase linked to Fanconi anemia and frequently mutated in cancers, counteracts replication stress by dismantling unconventional DNA secondary structures (such as G-quadruplexes) that occur at the DNA replication fork in certain sequence contexts. However, how FANCJ is recruited to the replisome is unknown. Here, we report that FANCJ directly binds to AND-1 (the vertebrate ortholog of budding yeast Ctf4), a homo-trimeric protein adaptor that connects the CDC45/MCM2-7/GINS replicative DNA helicase with DNA polymerase α and several other factors at DNA replication forks. The interaction between FANCJ and AND-1 requires the integrity of an evolutionarily conserved Ctf4-interacting protein (CIP) box located between the FANCJ helicase motifs IV and V. Disruption of the CIP box significantly reduces FANCJ association with the replisome, causing enhanced DNA damage, decreased replication fork recovery and fork asymmetry in cells unchallenged or treated with Pyridostatin, a G-quadruplex-binder, or Mitomycin C, a DNA inter-strand cross-linking agent. Cancer-relevant FANCJ CIP box variants display reduced AND-1-binding and enhanced DNA damage, a finding that suggests their potential role in cancer predisposition.

**Keywords** FANCJ/BRIP1; AND-1/WDHD1; DNA Replication; Genome Stability; G-quadruplex DNA
**Subject Category** DNA Replication, Recombination & Repair

## Introduction

Faithful DNA replication is critical for genomic stability maintenance. The DNA replication machinery is continuously challenged by damaged DNA templates and other obstacles (alternate secondary structures, R-loops, tightly bound proteins) present throughout the genome. All these physical barriers, which impede a smooth progression of the replication forks, give rise to the so-called replication stress (Saxena and Zou, 2022). This condition is also caused by nucleotide pool depletion or imbalance of replicative factor levels. DNA helicases are important players in counteracting replication stress due to their ability to resolve DNA secondary structures and/or remodel nucleic acid molecules during DNA repair/recombination reactions. The importance of these enzymes in maintaining genome homeostasis is proven by the fact that many of them are genetically linked to rare hereditary diseases characterized by genome instability, chromosome anomalies, developmental defects, and cancer predisposition.

FANCJ, also known as BRIP1 (for BRCA1-interacting protein 1) or BACH1 (BRCA1-associated C-terminal helicase 1), belongs to super-family 2 (SF2) Iron–Sulfur (Fe–S) cluster-containing DNA helicases, and is frequently mutated in breast and ovarian cancers, as well as in many other tumor types (Brosh and Cantor, 2014; Cantor et al, 2001). FANCJ belongs to the Fanconi anemia pathway and bi-allelic mutations of the encoding gene are known to cause the disease, which is characterized by hematopoietic stem cell defects, progressive bone marrow failure, genomic instability and cancer predisposition (Bridge et al, 2005; Levran et al, 2005; Levitus et al, 2005; Litman et al, 2005). Human FANCJ-deficient cell lines display increased sensitivity to Mitomycin C (MMC), a genotoxic agent that introduces DNA inter-strand cross-links (ICLs). FANCJ helicase activity and direct interaction with the mismatch repair protein MLH1 are both required for processing DNA ICLs (Peng et al, 2007). Nonetheless, the precise role of FANCJ in DNA ICL repair reactions remains poorly defined. In addition to BRCA1 and MLH1, other identified FANCJ-binding partners include the DNA exo/endonuclease MRE11 (Suhasini et al, 2013), the single-stranded DNA-binding replication protein A (RPA) (Sommers et al, 2014), the Bloom DNA helicase (BLM) (Suhasini et al, 2011), the translesion synthesis DNA polymerase REV1 (Lowran et al, 2020), the Topoisomerase II-binding protein TOPBP1 (Gong et al, 2010) and the DNA-end processing nuclease CtIP (Nath and Nagaraju, 2020). These multiple interactions reveal the involvement of FANCJ in various DNA repair pathways as well as in S-phase checkpoint activation.

Biochemical studies showed that the purified recombinant FANCJ protein has an ATPase-dependent DNA helicase activity with a 5' to 3' directionality that unwinds different DNA substrates in vitro, including DNA duplexes forming a fork or containing a 5'-flap, three-stranded displacement loops (D-loops) and various kinds of G-quadruplex (G4) DNA structures (Gupta et al, 2005;

[1]Istituto di Biochimica e Biologia Cellulare, Consiglio Nazionale delle Ricerche, Naples, Italy. [2]Università degli Studi della Campania "Luigi Vanvitelli", Caserta, Italy. [3]Structural Biology Laboratory, Elettra–Sincrotrone Trieste, Trieste, Italy. [4]IFOM ETS—The AIRC Institute of Molecular Oncology, Milan, Italy. [5]Istituto di Genetica Molecolare, Consiglio Nazionale delle Ricerche, Pavia, Italy. ✉E-mail: francesca.pisani@ibbc.cnr.it

London et al, 2008; Wu and Brosh 2009). Studies carried out in different systems [worms (Cheung et al, 2002), chicken DT40 (Sarkies et al, 2012) and human cells (Bharti et al, 2013) and *Xenopus laevis* cell-free egg extracts (Castillo Bosch et al, 2014; Sato et al, 2021)] have pointed toward a prominent role of FANCJ in G4 DNA cellular metabolism. Moreover, *FANCJ*-knockout (KO) mouse embryonic fibroblasts show increased microsatellite instability, a phenotype that is further exacerbated upon treatment with compounds that induce replication stress (Matsuzaki et al, 2015; Barthelemy et al, 2016). All the above considered, it was proposed that FANCJ plays a key role in dismantling DNA secondary structures and unconventional conformations (such as G4) that occur in certain sequence contexts, especially in transiently exposed single-stranded regions. Thus, FANCJ would prevent DNA double-strand break formation and ensure a smooth progression of the DNA replication machineries in stressful conditions (Brosh and Wu, 2021). Moreover, it has been recently demonstrated that FANCJ promotes the repair of DNA-protein cross-links by actively unfolding the protein adduct thereby, enabling its cleavage by the SPARTAN protease (Yaneva et al, 2023). Notwithstanding the biological significance of FANCJ association with the DNA replication machinery, the protein factor responsible for recruiting FANCJ at the replication forks has not been yet identified.

Human acidic nucleoplasmic DNA-binding protein 1 (AND-1), also known as WD repeat and high mobility group (HMG)-box DNA-binding protein 1 (WDHD1), is an adaptor protein, crucial for DNA replication, with orthologs in metazoans and in fungi. It was originally identified in a *Saccharomyces cerevisiae* genetic screen of mutants with an increased rate of chromosome loss and was named *Ctf4* for chromosome transmission fidelity 4 (Spencer et al, 1990). Both budding yeast Ctf4 and human AND-1 were demonstrated to bridge the CDC45/MCM2-7/GINS (CMG) replicative DNA helicase with DNA polymerase α, within the replication machinery (Gambus et al, 2009; Tanaka et al, 2009; Simon et al, 2014; Kilkenny et al, 2017; Guan et al, 2017). In budding yeast, Ctf4 loss has pleiotropic effects, including increased sensitivity to genotoxic agents, defective sister chromatid cohesion and modifications in the ribosomal DNA genomic *locus* (Villa et al, 2016; Samora et al, 2016; Fumasoni et al, 2015; Sasaki and Kobayashi, 2017). Instead, the Ctf4 orthologous proteins from *Schizosaccharomyces pombe* (Williams and McIntosh, 2005), *Drosophila* (Gosnell and Christensen, 2011) and chicken (DT40 cell system) (Abe et al, 2018) are essential for cell proliferation. In human cells, AND-1 depletion using siRNAs slows down cell cycle progression (Yoshizawa-Sugata and Masai, 2009). Subsequent structural studies revealed that budding yeast Ctf4 and human AND-1 share a similar multi-domain organization, each comprising a β-propeller (WD40) and a SepB module (Simon et al, 2014; Kilkenny et al, 2017; Guan et al, 2017). However, the human AND-1 polypeptide chain has a C-terminal extension that includes an HMG DNA-binding domain, not present in budding yeast Ctf4. The SepB domain is responsible for AND-1/Ctf4 trimerization and for interaction with proteins containing the so-called Ctf4-interacting protein (CIP) box. This is a conserved short peptide sequence, originally identified in the *S. cerevisiae* DNA polymerase α catalytic subunit and in GINS Sld5, which contains conserved acidic and hydrophobic amino-acid residues essential for interacting with Ctf4 (Simon et al, 2014). Other Ctf4-binding partners were identified in budding yeast, such as the helicase–nuclease Dna2, the

ribosomal DNA compaction protein Tof2, the replication initiation factor Dpb2 and the sister chromatid cohesion DNA helicase Chl1 (Simon et al, 2014; Villa et al, 2016; Samora et al, 2016). These findings suggest that Ctf4 is an interaction hub connecting the DNA replication machinery to multiple proteins and enzymes that contribute to diverse aspects of genome duplication.

Here, we report that human FANCJ directly interacts with AND-1 through a conserved CIP box that lies between the DNA helicase motifs IV and V, equivalent to the Chl1 CIP box. The FANCJ/AND-1 interaction is critical for recruiting FANCJ to the replisome and for promoting smooth progression of the DNA replication forks. We show that faulty association of FANCJ to AND-1, due to CIP box mutations, gives rise to increased DNA damage, replication fork stalling, and asymmetry in either unperturbed or stressful conditions. Besides, we report evidence that cancer-associated *FANCJ* mutant alleles causing substitutions of CIP box amino-acid residues are defective in AND-1-binding and induce enhanced DNA damage in normal and perturbed conditions. These findings provide important clues on how to interpret cancer risk and devise novel therapeutic strategies.

## Results

### Identification of a AND-1/Ctf4-interacting protein (CIP) box in FANCJ

The *S. cerevisiae* Chl1 DNA helicase was demonstrated to be recruited to the DNA replication forks through direct interaction with Ctf4. This interaction requires the integrity of a CIP box that is located between the conserved helicase motifs IV and V of the Chl1 polypeptide chain (Samora et al, 2016). Human DDX11 and FANCJ are paralogous proteins, both sharing sequence similarity with budding yeast Chl1. We identified a putative CIP box in the human FANCJ polypeptide chain between the helicase motifs IV and V by bioinformatic analysis. Other vertebrate FANCJ orthologs contain a similar putative CIP box that is located in the same position of their polypeptide chain (between the helicase motifs IV and V) and includes the amino-acid residues reported to be critical for AND-1/Ctf4-binding in other proteins (Fig. 1A) (Villa et al, 2016; Guan et al, 2017; Kilkenny et al, 2017). A multiple sequence alignment of the CIP boxes from various AND-1/Ctf4 client proteins is reported in Fig. 1A. In contrast, this putative CIP box is only partially conserved in human DDX11 or absent in other human Fe–S cluster DNA helicases (RTEL1 and ERCC2/XPD; Fig. EV1A). A model of the human FANCJ three-dimensional structure, based on an artificial intelligence prediction (Jumper et al, 2021), revealed that the putative CIP box is likely to fold as an α-helix and has a similar structure, position, and orientation as the budding yeast Chl1 CIP box (Fig. 1B).

These findings prompted us to test if FANCJ was indeed able to interact with AND-1. To this end, we carried out co-immunoprecipitation experiments on the nuclear fraction of S-phase-enriched HEK 293T cells and found that endogenous FANCJ and AND-1 are associated in cell nuclear extracts (Fig. 1C). We substituted highly conserved amino-acid residues of the putative FANCJ CIP box with Alanine to generate a FANCJ mutant named FANCJ AALA (Fig. 1A). HEK 293T cells were

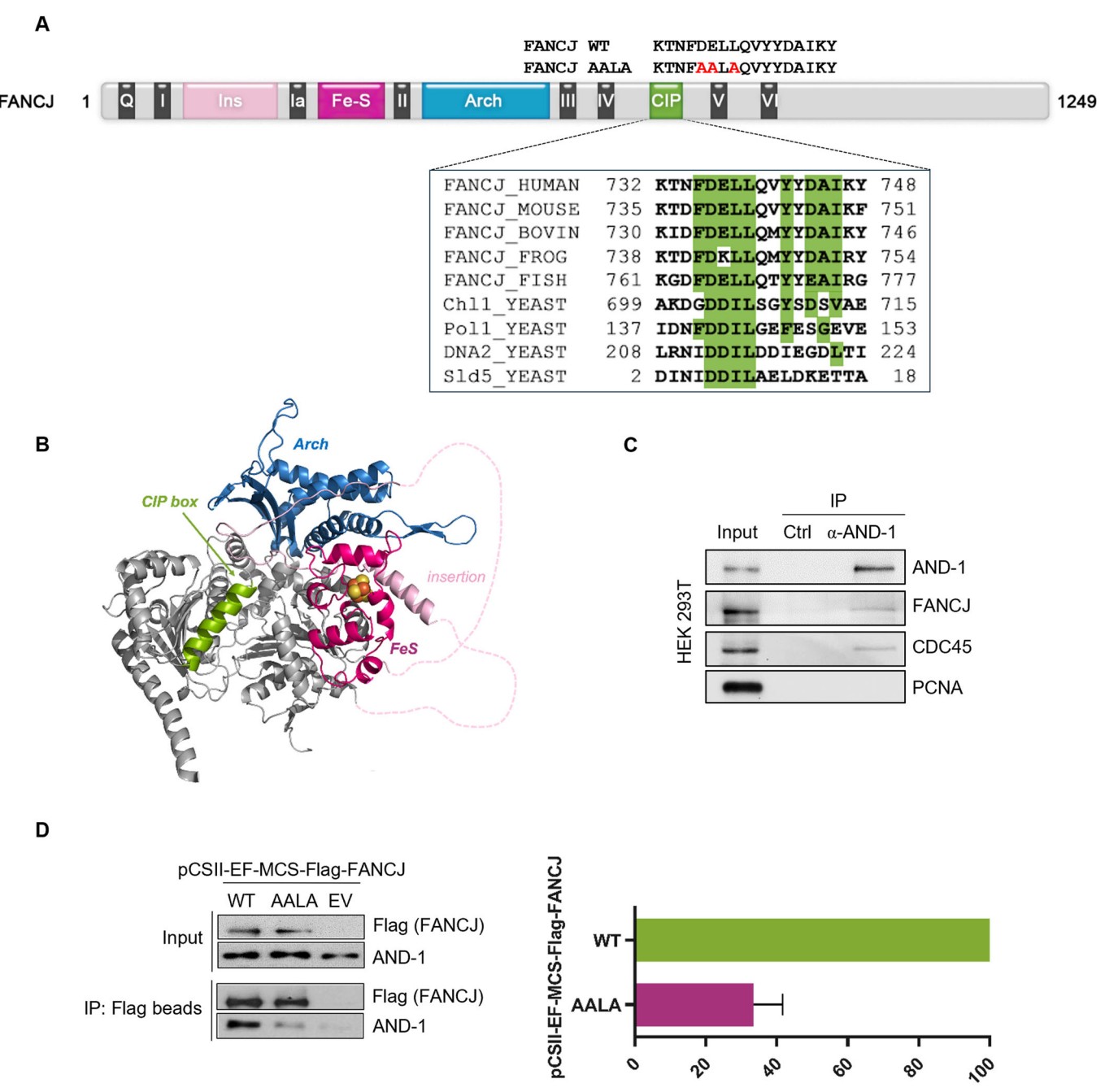

transiently transfected with plasmid vectors expressing Flag-tagged FANCJ wild type (WT) or the AALA mutant (AALA) and co-immunoprecipitation experiments were performed with anti-Flag M2 beads in whole-cell extracts. The results revealed that ectopically expressed Flag-tagged FANCJ WT was pulled down together with the endogenous AND-1 protein (Fig. 1D). In contrast, the amount of AND-1 co-immunoprecipitated with the Flag-tagged FANCJ AALA mutant was reduced by about 75%. All together, these results suggest that FANCJ associates with AND-1 in cell extracts, and this association is mainly dependent on the integrity of the newly identified FANCJ CIP box.

Then, we purified Flag-tagged FANCJ WT and AALA mutant proteins from transiently transfected HEK 293T cells (Fig. EV2A) and measured their DNA helicase activity using fluorescent-labeled DNA substrates in gel-based assays. We found that FANCJ AALA retained the same level of DNA helicase activity as FANCJ WT on an anti-parallel bi-molecular G4 DNA substrate (Fig. 7B), while it was able to unwind a forked duplex DNA as efficiently as the wild-type protein only at the highest concentrations tested (Fig. EV2B). In parallel control assays, FANCJ K52R, an ATPase-dead mutant, was found to have a very low level of DNA unwinding activity, which could be ascribed to a residual catalytic function observed

Figure 1. Identification of an AND-1/Ctf4-interacting protein (CIP) box in FANCJ.

(A) Schematic representation of the polypeptide chain of human FANCJ. Conserved helicase motifs (from I to VI) are indicated in dark gray (for simplicity motifs Va and Vb are not displayed). The identified CIP box is depicted in green. Other sequence motifs and specific domains are indicated with different colors, using the abbreviations: Q Q motif, Ins Insertion, Fe–S Fe–S cluster, Arch Arch domain. A multiple sequence alignment of FANCJ and other AND-1/Ctf4-interacting proteins CIP boxes is reported in the inset. The KALIGN tool (version 3.3.1) was used. Identical or highly similar amino-acid residues are highlighted in green, if present in most of the aligned sequences. The following abbreviations are used: HUMAN Homo sapiens, MOUSE Mus musculus, BOVIN Bos taurus, FROG Xenopus laevis, FISH Danio rerio, YEAST Saccharomyces cerevisiae, Chl1 chromosome loss 1, Pol1 DNA polymerase α catalytic subunit, Sld5 synthetic lethality with dpb11-1 protein 5. Changes of amino-acid residues to generate the FANCJ AALA mutant are indicated in red above the diagram. (B) Homology model of the FANCJ protein, as predicted by the AlphaFold server (Jumper et al, 2021), illustrating the location of the indicated conserved domains. The color code is the same as in (A). Disordered regions within the insertion are indicated by dashed lines; the Fe–S cluster was manually modeled based on the crystal structures of homologous DNA helicases. (C) Co-immunoprecipitation experiments were carried out with an anti-AND-1 antibody bound to Protein A Sepharose beads on the nuclear fraction of a S-phase-enriched HEK 293T cell population (lane indicated as IP). In a parallel control experiment, the anti-AND-1 antibody was omitted (lane indicated as Ctrl). Western blot analysis was carried out on the input (2% of each sample; 10 µg) and pulled down material (50% of each sample). Proteins of interest were detected using the indicated antibodies. PCNA was used as a negative control. The experiment was carried out in duplicate with consistent results. (D) Co-immunoprecipitation experiments were carried out with anti-Flag M2 beads on whole extracts from HEK 293T cells transfected with pCSII-EF-MCS vector, empty (EV) or expressing Flag-tagged FANCJ wild-type (WT) or the AALA mutant (AALA). Western blot analysis was carried out on the input (1% of each sample; 10 µg) and pulled-down material (50% of each sample). Proteins of interest were detected using the indicated antibodies. Co-immunoprecipitation experiments were done in triplicate. The level of immunoprecipitated AND-1 was normalized to pull down Flag-tagged FANCJ in each sample. Mean with standard errors is shown. Source data are available online for this figure.

only at high protein concentrations. Moreover, FANCJ K52R ATPase activity was almost completely abolished and not stimulated by DNA, as observed for the ATP hydrolysis reaction catalyzed by FANCJ WT and AALA mutant in side-by-side experiments (Fig. EV2C). Therefore, these results allowed us to ascribe the observed DNA unwinding activity to the purified recombinant FANCJ WT or the AALA mutant protein and not to any co-purified contaminant DNA helicase. In addition, these findings confirmed a separation-of-function between AND-1-binding and DNA unwinding for the FANCJ AALA mutant and allowed us to exclude the possibility that the CIP box mutations altered FANCJ protein structure and/or stability and thus the loss of AND-1-binding was simply due to protein misfolding.

## FANCJ directly interacts with the AND-1 SepB domain via its CIP box

Biochemical and structural studies revealed that human AND-1 and yeast Ctf4 share a similar three-dimensional structure, formed by a N-terminal WD40 domain and a SepB module (Fig. 2A), which is responsible for AND-1/Ctf4 trimerization and for binding CIP box-containing proteins. To examine whether there was a direct interaction between FANCJ and AND-1, we carried out in vitro pull-down assays using the purified recombinant full-length proteins (Fig. EV2A). Our analysis revealed that Myc-tagged AND-1, bound to Myc-agarose beads, was able to efficiently pull down FANCJ WT, but not the FANCJ AALA mutant (Fig. 2B). When the FANCJ DNA helicase activity was assayed in the presence of full-length AND-1, neither a stimulatory nor an inhibitory effect was observed on FANCJ ability to resolve either a G4 or a forked duplex DNA substrate in vitro.

We tested the ability of FANCJ to bind the AND-1 SepB domain, produced in bacterial cells as previously described (Kilkenny et al, 2017; Guan et al, 2017), in pull-down experiments using anti-Flag beads. As shown in Fig. 2C, we found that FANCJ WT pulled down 6xHis-SepB, while the FANCJ AALA mutant was almost completely unable to bind the AND-1 SepB domain.

We next expressed a short peptide containing the FANCJ CIP box, or its AALA mutant version, as a fusion with glutathione Stransferase (GST). We observed that 6xHis-SepB was pulled down by the GST-CIP box WT protein but not by its AALA mutant form

(Fig. 2D). Structural studies revealed that the SepB domain of either human AND-1 or budding yeast Ctf4 is made of a β-propeller followed by a bundle of 5 α-helixes (Simon et al, 2014; Kilkenny et al, 2017; Guan et al, 2017). The binding surface for the AND-1/Ctf4 client proteins consists of a groove between α-helices 2 and 4 (named H2 and H4) of the SepB helical bundle. The side chain of Methionine 766, which belongs to α-helix H2, is exposed on the groove surface and its substitution with an Alanine residue almost totally abolished the interaction of the SepB domain with the catalytic subunit of human DNA polymerase α, an AND-1-binding protein (Guan et al, 2017; Kilkenny et al, 2017). We produced the SepB α-helix bundle and its M766A mutant derivative as GST-fusion proteins (named GST-SepB α-helix bundle WT and M766A, respectively) and used these purified chimeric proteins to pull down FANCJ WT and the AALA mutant. As shown in Fig. 2E, a direct interaction was observed only between GST-SepB α-helix bundle WT (not the M766A mutant) and FANCJ WT (not the AALA mutant).

Collectively, these data demonstrate that FANCJ directly binds to AND-1 and this interaction requires the integrity of both the FANCJ CIP box and the AND-1 SepB α-helix bundle.

## FANCJ loss or its reduced AND-1-binding enhances DNA damage

To examine the physiological relevance of the FANCJ/AND-1 interaction, we carried out complementation assays in cells lacking the endogenous FANCJ protein and ectopically expressing FANCJ WT allele or its AALA mutant derivative. To this end, we generated a HeLa cell line where FANCJ was knocked out (FJ-KO) using CRISPR-paired guide RNAs targeting exons 2 and 3 (Fig. EV3A). Western blot analysis and Sanger sequencing of selected clones were carried out to confirm FANCJ protein loss and identify specific mutations introduced by the Cas9 activity. Since FANCJ depletion is expected to sensitize cells to Mitomycin C (MMC), a DNA inter-strand cross-linking agent, and Pyridostatin (PDS), a G4 DNA-stabilizer, we examined the viability of the FJ-KO HeLa cells after 5-day treatment with increasing doses of these drugs. As shown in Fig. EV3B, the FJ-KO HeLa cell line was sensitive to either MMC or PDS (with MMC exerting a toxic effect even in a low nM concentration range, 10–20 nM) and

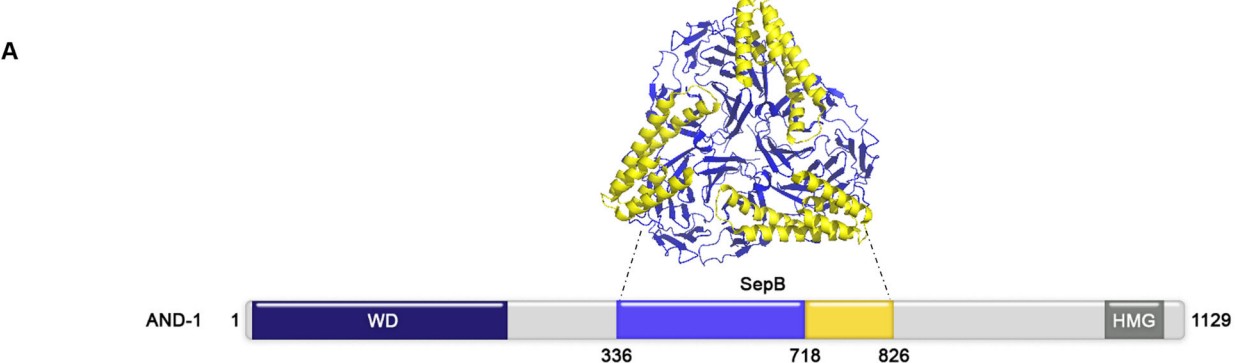

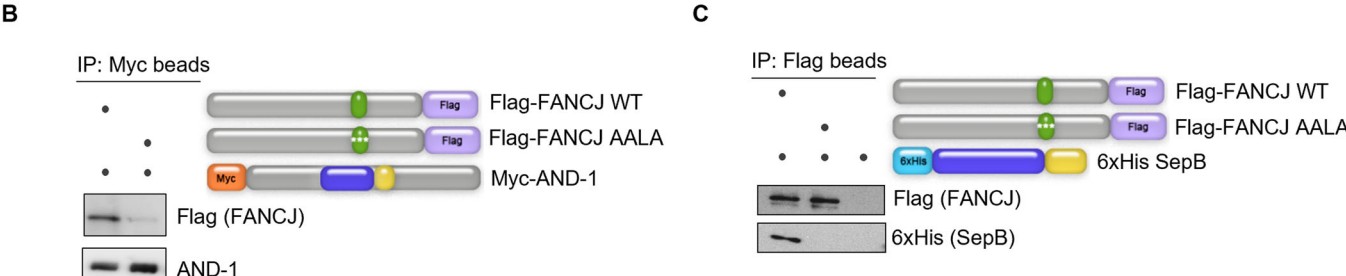

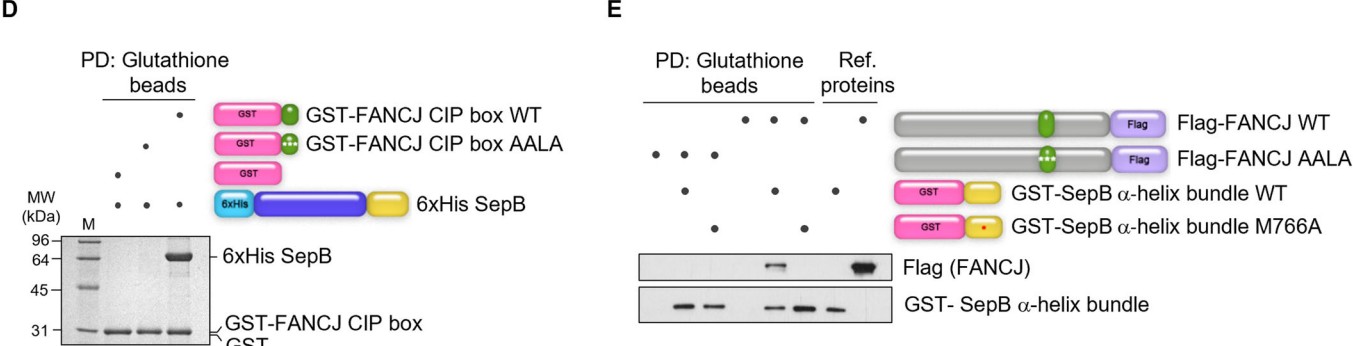

**Figure 2. FANCJ directly interacts with AND-1.**

(A) Schematic representation of the polypeptide chain of human AND-1. The WD40 repeat (*WD*), SepB (*SepB*), and high mobility group (*HMG*) domains are indicated by boxes of different colors. The β-propeller subdomain and α-helix bundle that form the SepB domain are depicted in purple and yellow, respectively. The same colors are used to highlight them in the human AND-1 SepB structure (Protein Data Bank ID 5gvb). (B) Pull-down experiments of Myc-tagged AND-1 full-length and Flag-FANCJ WT or AALA mutant using anti-Myc beads. Pulled-down samples (50% of their total volume) were analyzed by Western blot with the indicated antibodies. (C) Pull-down experiments of Flag-tagged FANCJ WT or AALA mutant and AND-1 6xHis-tagged SepB using anti-Flag M2 beads. Pulled-down samples (50% of the total volume for each blot) were analyzed by Western blot with the indicated antibodies. (D) GST was fused to the FANCJ CIP box (amino-acid residues 730–747) for use in a pull-down (*PD*) assay with purified AND-1 6xHis-SepB. Specific mutations of the FANCJ CIP box (as in the GST-FANCJ CIP box AALA derivative) disrupt its ability to pull down the SepB domain. (E) The GST-SepB α-helix bundle chimeric protein was able to pull down purified Flag-tagged FANCJ WT. Substitution of the critical Met766 with Ala in the SepB α-helix bundle abolishes FANCJ-binding. GST-pull-down assays were carried out with the indicated recombinant proteins. Pulled-down samples (50% of their total volume) were analyzed by Western blot with the indicated antibodies. Source data are available online for this figure.

re-expressing FANCJ WT or the AALA mutant in the FJ-KO line did almost completely revert sensitivity to both drugs.

We then stably complemented the FJ-KO HeLa line by expressing Flag-tagged FANCJ WT or the AALA mutant under the control of a Tetracycline-responsive promoter using the pMK240 plasmid vector (Okumura et al, 2018). Western blot analyses revealed that FANCJ WT and the AALA mutant were expressed at a comparable level and only upon Doxycycline

induction in the selected clones (Fig. EV3C). These stably complemented cell lines were used to analyze the impact of the FANCJ/AND-1 interaction on DNA damage induction following treatment with MMC and PDS. Analysis of γ-H2A.X *foci* by indirect immunofluorescence experiments revealed a higher level of DNA damage in FJ-KO compared to control HeLa cells, not only after treatment with MMC and PDS, but also in unperturbed conditions (Fig. 3). Moreover, complementation with the *FANCJ*

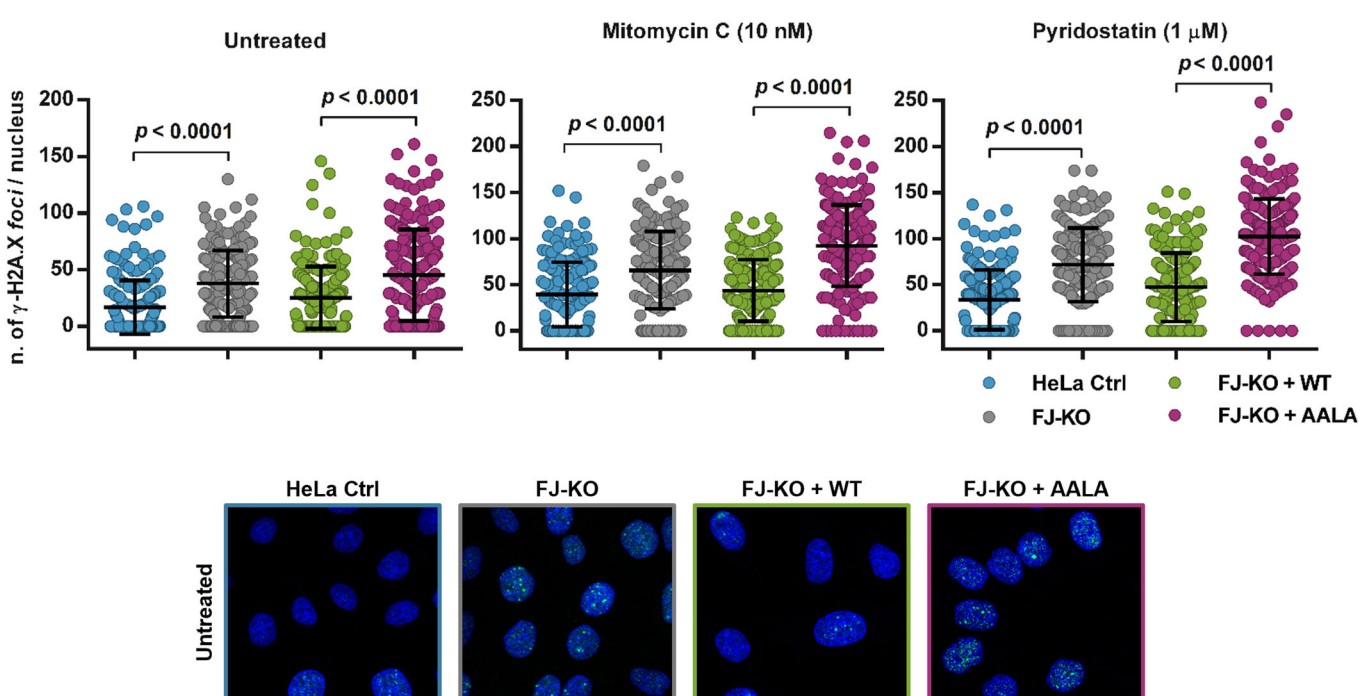

**Figure 3. FANCJ loss or reduced binding to AND-1 enhance DNA damage.**

The indicated cell lines were treated with PDS or MMC, after Doxycycline induction. γ-H2A.X *focus* formation was detected by immunofluorescence with a monoclonal antibody that specifically recognizes the Ser139-phosphorylated form of the above histone. Scale bar, 10 μm. Dot plot of the number of *foci* detected per cell is reported in each graph. Bars indicate mean ± SD; 200 cells were analyzed per condition; $n = 2$ biologically independent experiments, with at least two technical replicates per experiment; two-tailed $P$ value ($P < 0.01$) was calculated using Student's $t$ test nonparametric for unpaired data with Welch correction. Source data are available online for this figure.

WT allele almost completely reverted the observed phenotype, while expression of the *FANCJ* AALA variant gave rise to an even higher level of γ-H2A.X *foci* compared to the FJ-KO cells (either with or without drug treatment; Fig. 3).

These results indicate that the FANCJ/AND-1 interaction is important for suppressing DNA damage in unchallenged conditions or after treatment with MMC or PDS. However, re-expression of either FANCJ WT or the AALA mutant rescued the ability of the FJ-KO line to survive after exposure to both drugs. These findings indicate that FANCJ interaction with AND-1 is not critical for cell

viability after treatment with MMC or PDS and suggest that alternative DNA repair pathways are activated, albeit with different kinetics, to counteract enhanced DNA damage deriving from the faulty association of FANCJ to DNA replication forks.

## FANCJ associates with the DNA replication machinery through a direct interaction with AND-1

Given that FANCJ directly binds to AND-1 *via* its CIP box and AND-1 is a component of the DNA replication machinery, we asked whether

FANCJ localizes at DNA replication forks. Co-immunoprecipitation experiments were carried out using an antibody specific for CDC45, a component of the CMG complex, to identify proteins associated with the DNA replication machinery in cell extracts. HEK 293T cells, transiently transfected with plasmids expressing Flag-tagged FANCJ WT or the AALA mutant, were released in S phase after an overnight thymidine block. The nuclear fraction was prepared from these cells using a procedure that included a sonication step followed by an exhaustive nuclease treatment to remove nucleic acids that could mediate protein-protein associations. Co-immunoprecipitations were performed in the nuclear fraction samples using an anti-CDC45 antibody conjugated to Protein A Sepharose beads. As reported in Fig. 4A, Western blot analyses revealed that FANCJ WT was pulled down with CDC45, together with AND-1 and MCM4, a subunit of the MCM2-7 complex; in contrast, the amount of FANCJ AALA mutant co-immunoprecipitated with CDC45 was reduced by ~75% in experiment replicates. Since CDC45 is loaded onto chromatin only in S phase as a stable component of the CMG complex (Masai et al, 2010), our results revealed that FANCJ was anchored to the DNA replication machinery via a direct interaction with AND-1, as this interaction was dependent on the integrity of the FANCJ CIP box.

Thereafter, we examined the localization of FANCJ to DNA synthesis sites marked by EdU incorporation in FJ-KO HeLa cell lines complemented with FANCJ WT or AALA mutant allele using the in situ visualization of protein interactions at DNA replication forks (SiRF) technique (Roy et al, 2018). Control experiments revealed a comparable level of EdU-EdU proximity ligation assay (PLA) spots in all the cell lines tested, meaning that they had a similar number of active replication factories and an equal probability to produce a positive signal in the conditions used for the SiRF assay (Fig. 4B). Our analyses revealed the presence of robust FANCJ-EdU PLA signals either in HeLa control cells or in the FJ-KO line complemented with the FANCJ WT allele, indicating that FANCJ does co-localize with DNA synthesis sites. In contrast, FANCJ-EdU PLA spots were almost totally absent either in FJ-KO cells or in the FJ-KO line complemented with the FANCJ AALA mutant, suggesting that the CIP box AALA mutation significantly reduced the level of FANCJ associated to the DNA replication factories (Fig. 4B).

Collectively, these results reveal that FANCJ is associated with the DNA replication machinery via a direct interaction with AND-1. Our findings are consistent with a previous study, where nascent chromatin capture (NCC), used to profile chromatin proteome dynamics during S phase in human cells, indicated that FANCJ is among the proteins that are enriched at the replication forks in nascent chromatin (Alabert et al, 2014). Besides, in a more recent report the isolation of proteins on nascent DNA (iPOND) technique revealed the presence of FANCJ at replication forks in unchallenged conditions (Peng et al, 2018). However, although all these data would suggest that FANCJ is a constitutive component of the DNA replication machinery, the possibility that its recruitment at the replication forks is induced by endogenous DNA damage or replication fork barriers cannot be ruled out.

## AND-1-binding by FANCJ promotes replication fork progression in stressful conditions

Next, we examined the effect of disrupting the FANCJ CIP box on the replication fork dynamics using DNA fiber track assays to monitor DNA replication at a single-molecule level in the aforementioned cell lines. As schematically depicted in Fig. 5A, cells were first labeled with CldU and then a second label, IdU, was administered without or with PDS to test replication fork progression in normal or challenging conditions, respectively. A significant reduction of replication fork speed was measured upon PDS treatment in FJ-KO cells and in FJ-KO cells complemented with the AALA mutant, with a more prominent effect observed in this latter cell line (Fig. 5A). Analysis of sister forks (forks emanating from the same replication origin), obtained in PDS-treated conditions, revealed that the percentage of those with asymmetric tracks was remarkably higher in either FJ-KO cells or in FJ-KO cells complemented with the AALA mutant, indicating that replication forks stalled more frequently when FANCJ is either absent or not stably anchored to the ongoing replisomes (Fig. 5B).

## Cancer-relevant FANCJ CIP box mutations reduce AND-1-binding and induce enhanced DNA damage

After BRCA1 and BRCA2, FANCJ is the third most common ovarian cancer susceptibility gene: nearly 0.9– 2.5% of all ovarian cancer patients carry a splice-site, stop, or frameshift mutation in the FANCJ gene (Ramus et al, 2015; Norquist et al, 2016). Besides, FANCJ was found to be mutated in several other malignancies, including melanoma, breast, prostate, and hereditary colon cancer, suggesting that FANCJ mutations may be a risk factor in multiple tumor types (Cantor and Guillemette, 2011; Paulo et al, 2018; Ali et al, 2019). Nevertheless, a direct association of FANCJ mutations with predisposition to breast cancer has been questioned (Weber-Lassalle et al, 2018; Easton et al, 2016). However, as the majority of FANCJ clinical variants remain uncharacterized, their possible connection to cancer risk is difficult to interpret (Moyer et al, 2020; Calvo et al, 2021). Through data mining of cancer genomics databases (cBioPortal, http://cbioportal.org, Cosmic-3D, https://cancer.sanger.ac.uk/cosmic3d, and gnomAD, https://gnomad.broadinstitute.org) we pinpointed FANCJ variants expected to give rise to mutations of CIP box amino-acid residues in various malignancies, including uterine endometrial carcinoma (N734H), invasive breast carcinoma (D736H), breast and ovarian cancer (Q740H), glioblastoma multiforme (Y743C), and colorectal adenocarcinoma (A745T) (Fig. 6A).

To test if the above FANCJ mutant derivatives were able to bind AND-1, we carried out co-immunoprecipitation experiments with anti-Flag agarose beads in whole extracts of FJ-KO HeLa cells transiently transfected with plasmids expressing Flag-tagged versions of the FANCJ mutants of interest. As reported in Fig. 6B, the FANCJ CIP box mutants N734H and D736H displayed a reproducibly reduced association with the endogenous AND-1 protein, as also found for the FANCJ AALA mutant.

The FANCJ N734H and D736H mutants were produced as Flag-tagged recombinant proteins and purified from transiently transfected HEK 293T cells (Fig. EV2A). Then, GST-pull-down assays were carried out to examine their direct interaction with the GST-SepB α-helix bundle chimeric protein. Results from these experiments confirmed that AND-1-binding by both these FANCJ mutants was reduced, even if not completely abolished (Fig. 7A). When assessing the activity towards a G4 DNA substrate, the FANCJ N734H and D736H mutants displayed the same level of unwinding as the wild-type protein (Fig. 7B). When tested towards

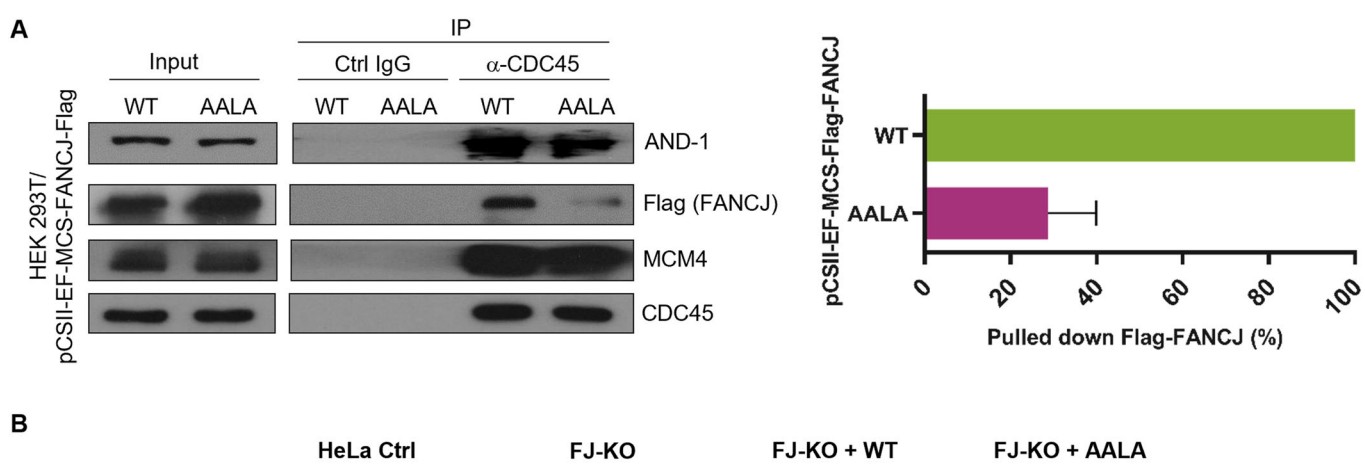

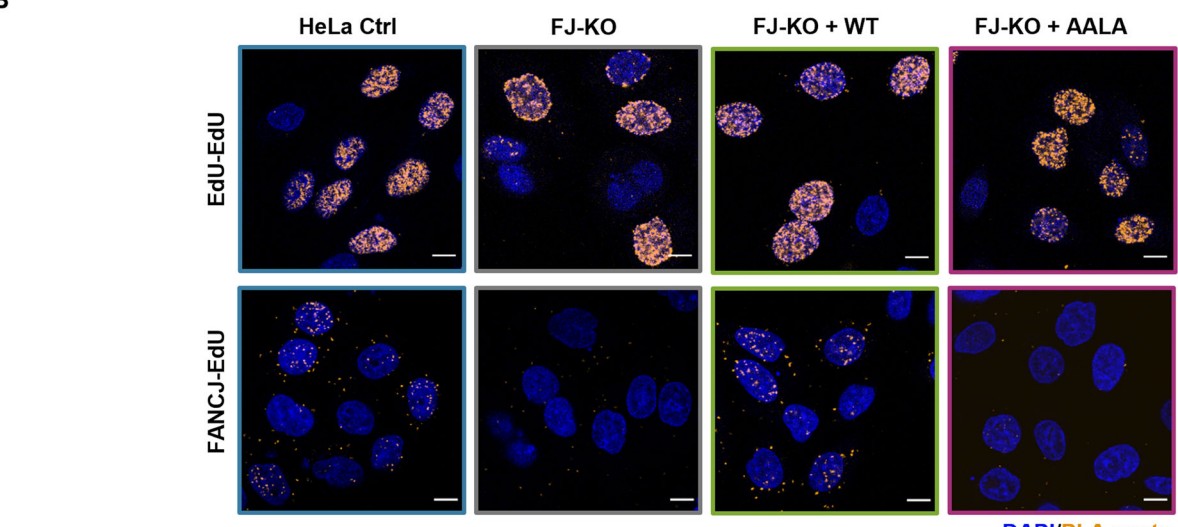

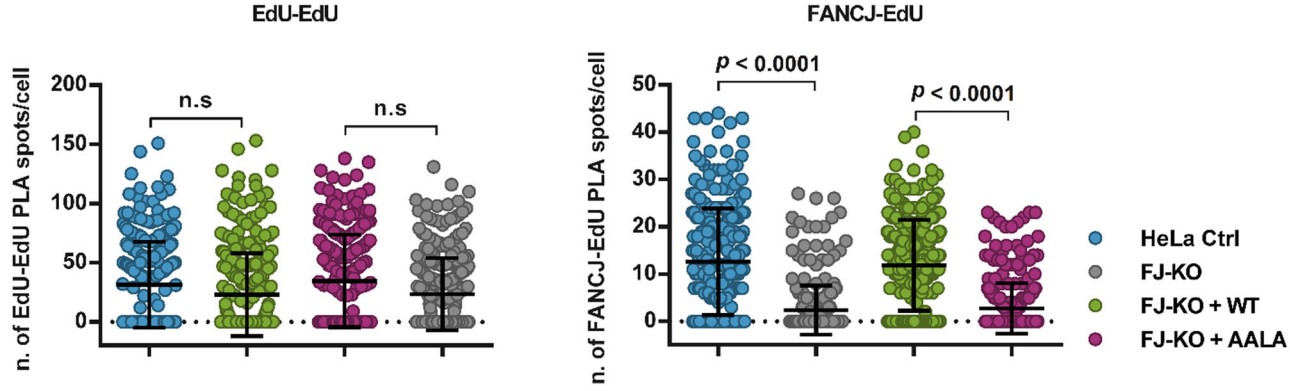

**Figure 4. FANCJ associates to DNA replication forks through a direct interaction with AND-1.**

(A) Co-immunoprecipitation experiments were carried out on the nuclear fraction of HEK 293T cells transiently transfected with pCSII-EF-MCS plasmid constructs expressing Flag-tagged FANCJ wild-type (*WT*) or AALA mutant (*AALA*). Control (*Ctrl IgG*) and anti-CDC45 rabbit IgG (α-*CDC45*), bound to Protein A Sepharose beads, were added to each indicated nuclear fraction. Western blot analysis was carried out on the input (2% of each sample; 10 μg) and pulled-down material (50% of each sample). Proteins of interest were detected using the indicated antibodies. Immunoprecipitated protein samples were analyzed by Western blot using the indicated antibodies. Data from triplicate experiments show level of immunoprecipitated FANCJ (mean ± SD), normalized to pulled-down endogenous CDC45, in each sample. Means with standard errors are shown. (B) FANCJ association to sites of DNA synthesis was analyzed by SiRF assays. EdU-EdU (upper part) or FANCJ-EdU (lower part) proximity ligation assay (PLA) spots were quantified from control (*Ctrl*), FJ-KO and complemented HeLa cell lines. Scale bar, 10 μm. The frequency distribution of the population was analyzed and two-tailed *P* value (*P* < 0.01) was calculated using Student's *t* test nonparametric for unpaired data with Welch correction (< 300 cells per condition; *n* = 3 biologically independent experiments; bars show mean ± SD). Source data are available online for this figure.

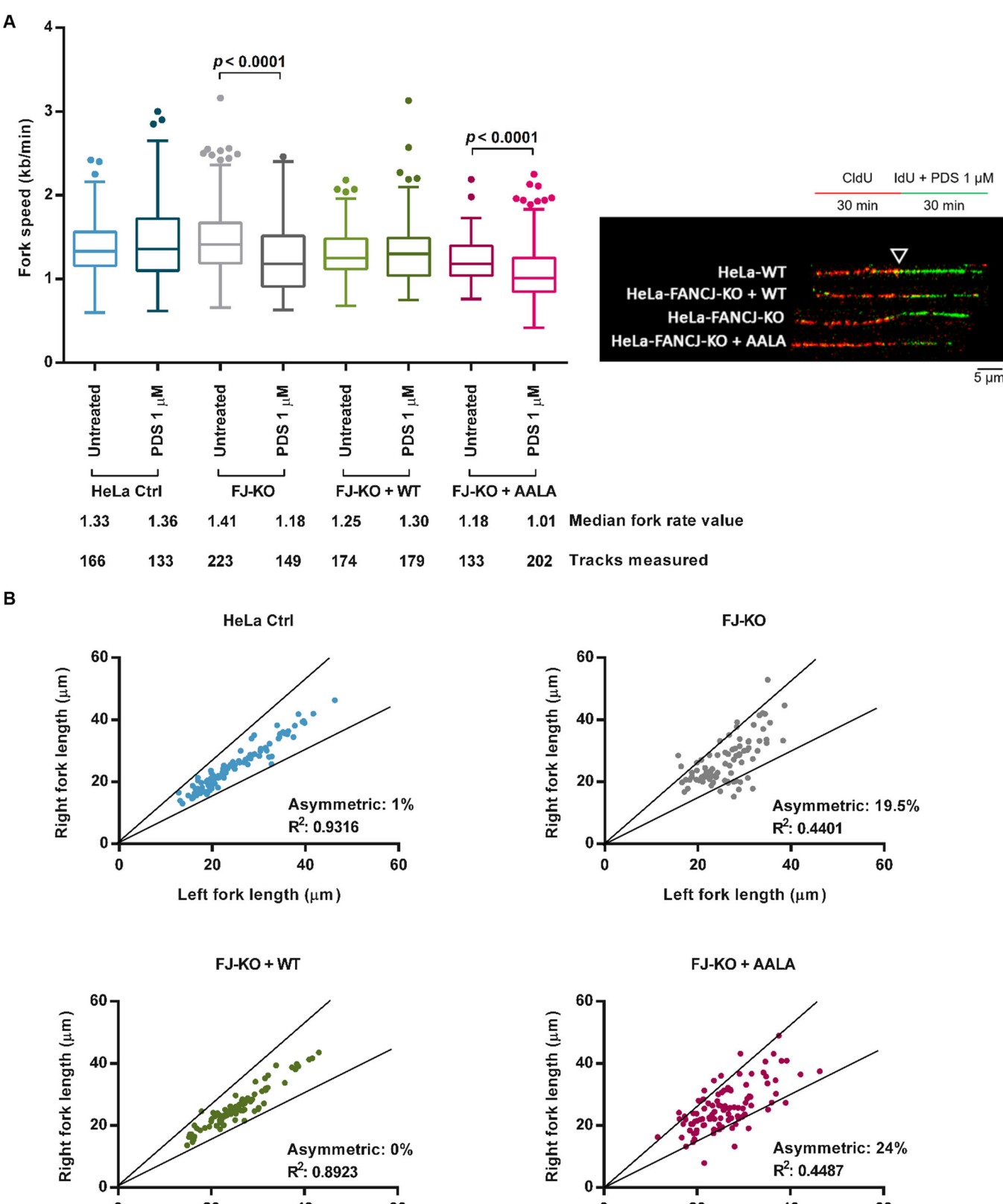

**Figure 5.   Association of FANCJ to the DNA replication machinery promotes fork progression in challenged conditions.**

(A) DNA fiber track analysis was carried out in either untreated or perturbed conditions. Box-plot showing fork speed determined on CldU:IdU double-labeled DNA fibers from control (*Ctrl*), FJ-KO and stably complemented HeLa cell lines, untreated or treated with PDS (1 μM) simultaneously with IdU pulse ($n = 3$ biologically independent experiments, with at least 2 technical replicates). Labeled DNA fibers are shown from a representative experiment (arrow represents the switch point from CldU to IdU labeling). Median fork rate and the number of tracks analyzed are shown. The box extends from the 25th minus 1.5×IQR to 75th plus 1.5×IQR percentiles—Tukey method. Two-tailed *P* values were calculated by Mann–Whitney *U* test. (B) Fork asymmetry analysis in cells treated with PDS during IdU pulse, as in (A). The central area between the lines delimits a variation of <25% in fiber length. $R^2$ is the linear correlation coefficient. Approximately 100 sister forks (green-red-green tracks only) were analyzed and plotted from three biologically independent experiments. Source data are available online for this figure.

a canonical forked duplex DNA, FANCJ N734H was as efficient as FANCJ WT, while the D736H mutant was as active as FANCJ WT on this substrate only at the highest concentrations tested (Fig. EV2B). In parallel control assays, when ATP was omitted from the assay, FANCJ WT did not display a detectable helicase activity on either a G4 or a forked duplex DNA substrate. In contrast, FANCJ K52R, an ATPase-dead mutant, was found to have a very low level of DNA unwinding, which could be ascribed to a residual catalytic function observed only at high protein concentrations, and not to another contaminating DNA helicase. In agreement with this hypothesis, FANCJ K52R was found to possess a very low level of ATPase activity that was not stimulated in the presence of DNA. In contrast, the ATPase activity of FANCJ WT and its mutants of interest was found to be highly enhanced in the presence of DNA in side-by-side experiments (Fig. EV2C). Therefore, our results confirmed previously published data indicating that FANCJ DNA helicase function is ATP-dependent and allowed us to ascribe the observed DNA unwinding activity to the purified recombinant FANCJ WT protein or its CIP box mutants of interest and not to any co-purified contaminant protein. Overall, these findings confirmed a separation-of-function between AND-1-binding and DNA helicase activity for the FANCJ N734H and D736H CIP box mutants, as also previously found for the FANCJ AALA mutant.

Moreover, we found that the above FANCJ CIP box mutants retain the ability to associate with BRCA1, BLM, and MLH1, other known FANCJ-binding partners in cell extracts (see Fig. EV4A), suggesting that their involvement in the related genome maintenance pathways is not affected by the above CIP box amino-acid substitutions.

Then, we examined the effect of the FANCJ N734H and D736H mutants on DNA damage induction in cells untreated/treated with MMC or PDS. This analysis revealed that the above FANCJ CIP box mutants were unable to counteract increased γ-H2A.X *focus* formation in FJ-KO HeLa cells (see Fig. EV4B).

## Discussion

Here, we report the identification of an AND-1/Ctf4-interacting protein (CIP) box in the human FANCJ polypeptide chain between the conserved DNA helicase motifs IV and V. We found that endogenous FANCJ and AND-1 are associated in cell extracts and demonstrated that a direct interaction takes place between the two proteins produced in purified recombinant form. Moreover, pull-down experiments with various AND-1 and FANCJ protein fragments revealed that the FANCJ CIP box directly binds to the α-helix bundle of the AND-1 SepB domain (Fig. 2). Notably, only some of the CIP box residues were found in the same region of the

DDX11 DNA helicase sequence and we were unable to detect any direct interaction between Flag-tagged DDX11 and AND-1 SepB in co-immunoprecipitations carried out in vitro with the purified recombinant proteins (Fig. EV1), in agreement with a previous report (Farina et al, 2008).

Whereas the majority of CIP boxes fold as short α-helices that are typically surrounded by partially disordered regions, the *S. cerevisiae* Chl1 CIP box is located on an α-helix that is an integral part of the structure, although reasonably exposed to the solvent. As previously pointed out, in an AlphaFold model of the human FANCJ structure, the proposed CIP box is exactly structured and positioned as the budding yeast Chl1. When compared with the CIP boxes present in low complexity sequences (such as those found in human Pol α and in budding yeast Sld5, Pol1, and DNA2), some of the residues that are expected to interact with AND-1/Ctf4 (i.e., FANCJ L738 and L739 and Chl1 I705 and L706) are partly buried. However, the CIP box α-helices of both FANCJ and Chl1 appear to be loosely associated with the bulk of helicase structure, and it is not impossible to conceive a small conformational change that would expose those residues to the solvent. Moreover, no experimental structure for either FANCJ or Chl1 is available, and our bioinformatic analyses rely on computational models that are designed to place hydrophobic residues toward the inside of the protein; the presence of conserved hydrophobic amino acids may therefore drive the algorithm to artificially bury those residues. Indeed, the interaction between budding yeast Chl1 and Ctf4 is supported by a growing body of evidence (Borges et al, 2013; Samora et al, 2016; Villa et al, 2016), strongly suggesting that the conformation of the Chl1 CIP box α-helix is not an impediment. Nevertheless, an alternative explanation would be that FANCJ (or Chl1) binds AND-1 (or Ctf4) in a different fashion involving other residues within the same region.

The physiological relevance of the FANCJ/AND-1 interaction was examined by functional assays carried out in FJ-KO HeLa cells stably complemented with either *FANCJ* WT or the AALA mutant allele. The FANCJ AALA derivative is a separation-of-function mutant as it is defective in AND-1-binding due to changes in critical CIP box residues but retains a level of DNA helicase activity comparable with the wild-type protein. Of note, we found that the CIP box mutations described in this study (AALA, N734H, and D736H) did not abolish the ability of FANCJ to associate with BRCA1, BLM, and MLH1, suggesting that the related FANCJ functions could be not altered by these amino-acid changes (see Fig. EV4A). Our data revealed that the integrity of the CIP box was critical for recruiting FANCJ to the replication factories and for suppressing DNA damage in either unperturbed or challenged conditions (treatment with MMC or PDS). In addition, we found that the FANCJ/AND-1 interaction protected DNA replication forks by promoting their progression and stability, especially in

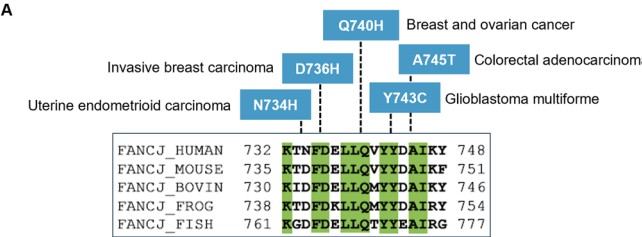

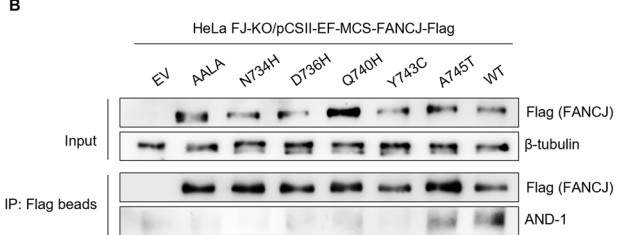

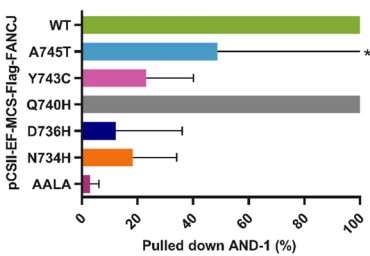

**Figure 6. Cancer-associated FANCJ CIP box mutations show reduced AND-1-binding in cells.**

(A) Cancer-relevant FANCJ CIP box mutants. Amino-acid changes corresponding to *FANCJ* missense variants found in the indicated tumors are shown in the upper part. A multiple sequence alignment of the CIP box from various FANCJ orthologs is reported in the lower part; highly conserved amino-acid residues are highlighted in green. The following abbreviations were used: *HUMAN Homo sapiens, MOUSE Mus musculus, BOVIN Bos taurus, FROG Xenopus laevis, FISH Danio rerio.* (B) Co-immunoprecipitation experiments with anti-Flag agarose beads were carried out on whole extracts of HeLa FJ-KO cells transiently transfected with pCSII-EF-MCS vector expressing Flag-tagged FANCJ wild type (*WT*) or the indicated mutants (*EV* stands for empty vector). Western blot analysis was carried out on the input (1% of each sample; 5 µg of protein) and pulled-down material (20 and 40% of each sample for FANCJ-Flag and AND-1 detection, respectively). Proteins of interest were detected using the indicated antibodies. Experiments were done in triplicate. The level of immunoprecipitated endogenous AND-1 was quantitated in each sample, as described in "Methods". Mean ± SD are shown in the bar graph, asterisk on A745T bar refers to a standard deviation of 58.44. Source data are available online for this figure.

cells treated with PDS. These findings are consistent with several lines of evidence suggesting that FANCJ has a prominent role in resolving G4 structures at DNA replication forks (Cheung et al, 2002; Wu et al, 2008; Schwab et al, 2013; Castillo Bosch et al, 2014; Sato et al, 2021; Van Schendel et al, 2021) and with biochemical studies showing that FANCJ is able to untangle different kinds of G4 DNA structures in vitro with higher catalytic efficiency, as compared to other DNA helicases (Bharti et al, 2013). Of note, G4 *foci* were reported to accumulate in FANCJ-deficient human

cells, revealing that other G4 resolvases only partially compensate for FANCJ loss (Henderson et al, 2014; Summers et al, 2021). Recent studies based on multi-color single-molecule localization microscopy have provided direct evidence that in human cells G4 are formed at DNA replication forks behind the CMG complex and resolved by the combined action of FANCJ and replication protein A (RPA) (Odermatt et al, 2020; Lee et al, 2021). In recent structural studies of the human CMG complex bound to AND-1 (Rzechorzek et al, 2020) or the whole human core replisome (Jones et al, 2021), the SepB trimer appears to be located near-perpendicularly to the N-tier face of the MCM2-7 complex, where it interfaces with CDC45 and GINS via its β-propeller domain, while the SepB α-helix bundle and the C-terminal HMG subdomain project away from the CMG, likely in close proximity to the lagging strand. Thus, FANCJ, bound to the SepB helical bundle, occupies a position in the replisome favorable to promptly hop onto the lagging strand between the Okazaki fragments and/or the unwound leading template as it emerges from the MCM complex, where G4 DNA structures are expected to arise. Notably, in a study carried out in DT40 cells it was suggested that G4 occurring at the replication forks could be resolved by the interplay of DDX11 and its binding partner TIMELESS, a component of the fork-protection complex (Lerner et al, 2020). In fact, in this cell system DDX11 and TIMELESS were found to act epistatically in preventing G4-dependent epigenetic instability, but independently from FANCJ. In agreement with these findings, herein we propose the existence of at least two partially redundant pathways responsible for G4 resolution at the DNA replication forks in human cells: one operating on the DDX11-TIMELESS axis and the other one involving FANCJ and its binding partner AND-1 (Calì et al, 2015; Cortone et al, 2018; Lerner et al, 2020).

Furthermore, our data suggest that the FANCJ/AND-1 interplay could also be relevant for repairing DNA ICLs. The integrity of the Fanconi anemia pathway is required to fix these DNA lesions and allow restart of stalled DNA replication forks. In a very recent work, it has been demonstrated that ATR-mediated phosphorylation of AND-1 at Threonine 826 (an amino-acid residue of the SepB helical bundle) promotes recruitment of the FANCM/FAAP24 complex at ICL-stalled DNA replication forks (Zhang et al, 2022). Of note, the authors of this study have proposed that AND-1 initiates the Fanconi anemia pathway by sensing and specifically binding DNA ICLs through its C-terminal HMG domain: threonine 826 phosphorylation would elicit an AND-1 conformational change that enhances affinity of the HMG domain for DNA ICLs promoting FANCM/FAAP24 recruitment to these damaged sites.

Our study also addresses the possible functional significance of two *FANCJ* missense variants (*FANCJ* N734H and D736H), that are found in tumor patients and result in mutations of CIP box amino-acid residues (Moyer et al, 2020; Calvo et al, 2021). We observed that AND-1-binding by these cancer-related FANCJ mutants is severely compromised, while the DNA helicase activity and the interaction with other known protein binding partners (BRCA1, BLM and MLH1; see Fig. EV4A) is not altered, as compared to the wild-type protein. Moreover, the FANCJ N734H and D736H mutants were found to be unable to counteract DNA damage induced by treatment with PDS or MMC in FJ-KO HeLa cells, likewise the FANCJ AALA mutant (see Fig. EV4B). Our

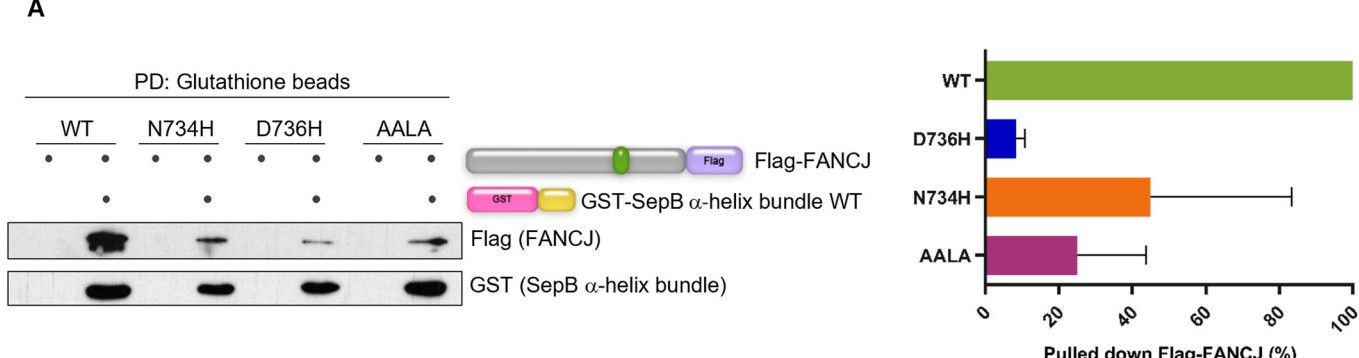

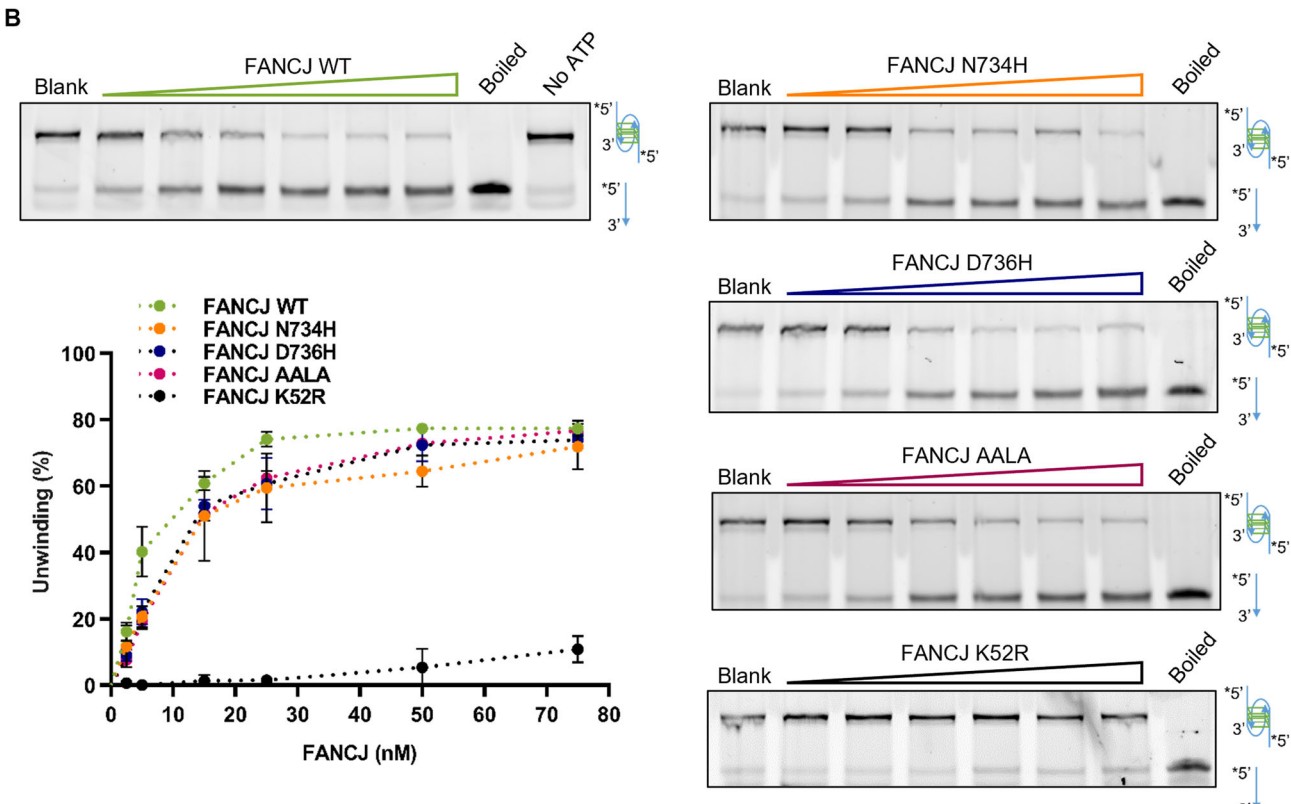

**Figure 7. Biochemical characterization of the cancer-related FANCJ CIP box mutants N734H and D736H.**

(A) GST-pull-down experiments of the GST-SepB α-helix bundle protein and Flag-tagged FANCJ WT, AALA, N734H, or D736H mutants. Reduced binding to SepB α-helix bundle is observed for all the FANCJ variants, compared to the wild-type protein. Mean ± SD are shown in the bar graph ($n = 3$ independent experiments). (B) Gel-based DNA helicase assays were carried out using FANCJ WT or the indicated mutants at increasing concentrations and a fluorescent-labeled anti-parallel bi-molecular G4 DNA substrate. Asterisk represents the fluorophore attached to the DNA oligonucleotide 5′ end. Blank and No ATP refer to control assays without protein or ATP, respectively. Boiled refers to a heat-denatured assay mixture with no protein. Graphs report data of three independent experiments (mean ± SD) carried out as described in the text. Source data are available online for this figure.

findings offer a possible interpretation of the functional significance of these clinically relevant *FANCJ* variants, showing that the association of FANCJ to the DNA replication machinery *via* AND-1 is crucial for preserving genomic integrity in either normal or perturbed conditions. Further analysis evaluating tumor frequency in *FANCJ*-knocked-in mouse models is required to assess impact and prognostic relevance of the above cancer-associated *FANCJ* variants.

# Methods

## Plasmid construction and protein expression and purification

Plasmid pCSII-EF-MCS (version 3.4) harboring the human Flag-tagged FANCJ open reading frame (ORF) was a gift from Hisao Masai (Tokyo Metropolitan Institute of Medical Science, Tokyo,

Japan) (Uno et al, 2012). The FANCJ mutant alleles, named AALA and N734H, D736H, Q740H, Y743C, A745T, and K52R, were produced by a PCR-mediated site-directed mutagenesis protocol (Carey et al, 2013) using the Q5 DNA polymerase (New England Biolabs Laboratories) or KOD Hot start (Merck). Plasmids (pCSII-EF-MCS version 3.4) expressing human Flag-tagged FANCJ WT and its mutant forms, named AALA, N734H, D736H, and K52R, were transfected into HEK 293 T cells grown on ten 15-cm dishes. After 48 h, cells were detached using ice-cold PBS (150 mM NaCl, 2.7 mM KCl, 10 mM Na$_2$HPO4, 1.8 mM KH$_2$PO4, pH 7.5) and collected by centrifugation (250× $g$, 3 min, 4 °C). Cell pellets were resuspended in Binding Buffer (50 mM Tris-HCl, pH 7.0, 150 mM NaCl, 10% [v:v] glycerol, 0.1% [v:v] Triton X-100 and 2× cOmplete EDTA-free protease inhibitor cocktail [Roche]). Cell lysis was obtained by a mechanical pestle and sonication, following digestion with Benzonase (Merck) at 25 units/mL (in the presence of 5 mM MgCl$_2$) for 30 min. The sample was ultracentrifuged (21500× $g$, 45 min, 4 °C), the supernatant filtered with 0.22-μm filter and mixed with anti-Flag M2 agarose beads (Merck) for 2 h in a rotating wheel at 4 °C. Resin was washed with Binding Buffer containing increasing amounts of NaCl (150 mM, 300 mM, and 500 mM). Bound protein was eluted from the resin using Elution Buffer (50 mM Tris-HCl, pH 7.0, 300 mM NaCl, 10% [v:v] glycerol, 0.1% [v:v] Triton X-100, 0.2 μg/μL 3× Flag-peptide, 2× cOmplete EDTA-free protease inhibitor cocktail [Roche]). The eluted sample was concentrated on a Vivaspin (cutoff: 10 kDa). Sample buffer was exchanged by diafiltration in Storage Buffer (25 mM Tris-HCl, pH: 7.5, 150 mM NaCl, 10% [v:v] glycerol, 1 mM PMSF, and 1 mM dithiothreitol). The final yield of the purified full-length proteins ranged from about 50 to 150 μg from a total of 9 × 10$^7$ HEK 293T-transfected cells.

The pFBDM vector plasmid harboring the ORF encoding full-length 10xHis-Myc-tagged AND-1 (amino-acid residues 1–1129; a gift of Luca Pellegrini, Cambridge, United Kingdom) was used to generate a recombinant baculovirus (Kilkenny et al, 2017). To express the AND-1 protein, High-Five™ insect cells, grown in suspension at a concentration of $1.0 \times 10^6$ cells/mL (culture volume: 1 L), were infected with a recombinant baculovirus stock and incubated for 72 hr at 27 °C. Then, cells were harvested by centrifugation (1000× $g$, 1 hr, 4 °C), washed with PBS supplemented with 10% (v:v) glycerol and stored at −80 °C. After thawing, cells were resuspended in ten volumes of Lysis Buffer (25 mM Tris-HCl, pH 7.2, 300 mM NaCl, 10% [v:v] glycerol, 0.5 mM Tris(2-carboxyethyl)phosphine [TCEP], 20 mM imidazole), complemented with protease inhibitors (cOmplete, EDTA-free) and incubated for 15 min on ice. Cells were disrupted by 5 × 10-s pulses of ultrasound at 35% amplitude interrupted by 45-s incubations on ice using a Branson 450 Digital Sonifier. Benzonase (Merck) was added at 2.40 units/mL and, after incubation at room temperature (RT) for 15 min, the crude extract was centrifuged (30000× $g$, 4 °C, 1 h). A 1-mL aliquot of Ni-NTA resin (Qiagen), pre-equilibrated with Lysis Buffer, was added to the supernatant. The mixture was incubated at 4 °C for 2 h. The resin was collected in a 20-mL column (bed volume: 2 mL) and washed with Lysis Buffer (volume: 30 mL). Resin was further washed with Lysis Buffer containing 5 mM MgCl$_2$, 0.5 mM ATP (10 column volume, CV), followed by an additional wash with Lysis Buffer (10 CV). The AND-1 protein was eluted in 5-mL fractions with Elution Buffer (25 mM Tris-HCl, pH 7.2, 300 mM NaCl, 10% [v:v] glycerol, 0.5 mM TCEP, 250 mM

imidazole). The sample was diluted in Elution Buffer without NaCl so that the salt concentration was reduced to 100 mM. Then, the sample was applied to a 1-mL HiTrap Q column (Cytiva), pre-equilibrated with anion exchange buffer (25 mM Tris-HCl pH, 7.2, 100 mM NaCl, 10% [v:v] glycerol, 0.5 mM TCEP). AND-1 was eluted with a 20-CV gradient from 150 to 1000 mM NaCl. Peak fractions were pooled and concentrated using an Amicon Ultra-15 system (cutoff: 30 kDa).

Recombinant human 6xHis-tagged AND-1 SepB (amino-acid residues 336–826) was produced in *Escherichia coli* BL21 (*DE3*) cells transformed with pRSF-Duet-1 plasmid construct (a gift from Luca Pellegrini, Cambridge, United Kingdom) and purified, as previously described (Kilkenny et al, 2017).

The FANCJ CIP box (amino-acid residues 730–747) wild-type (WT) and its AALA mutant version were produced as GST-fused proteins by cloning the encoding ORF sequences into the pGEX-6P-1 plasmid (GenScript).

The SepB α-helix bundle (amino-acid residues 718–824) was produced as a GST-fused protein by cloning the encoding ORF into the pGEX-2T plasmid. The recombinant chimeric protein was expressed in *E. coli* Rosetta p*LysS* (*DE3*) cells, grown in LB medium at 37 °C. When the A$_{600}$ of the culture reached 0.6 optical density (OD), the cell culture was cooled down at 20 °C, and isopropyl thio-D-galactopyranoside (IPTG) was added at 0.1 mM. After an overnight incubation at 20 °C, cells were collected by centrifugation and resuspended in Binding Buffer (PBS containing: 2× cOmplete EDTA-free protease inhibitor cocktail [Roche], 1 mg/mL Lysozyme, 25 units/mL Turbonuclease and 5 mM MgCl$_2$). Cells were broken by sonication and incubated for 20 min on ice, in a shaking platform, to allow further digestion of nucleic acids by Turbonuclease. Thereafter, NaCl concentration was adjusted to 300 mM and the sample was ultracentrifuged (21500 $g$, 45 min, 4 °C). The supernatant was filtered through a 0.22-μm filter and mixed with Glutathione Sepharose 4B agarose beads (Cytiva). The sample was incubated for 1 h in a rotating wheel at 4 °C. Resin was washed with Binding Buffer (PBS supplemented with 150 mM NaCl) and protein was eluted with Elution Buffer (50 mM Tris-HCl, pH 7.5, 300 mM NaCl and 10 mM reduced glutathione). The eluted sample was concentrated using a Vivaspin system (cutoff: 10 kDa). Sample buffer was exchanged by diafiltration in Storage Buffer (50 mM Tris-HCl, pH: 7.5, 300 mM NaCl, 1 mM PMSF and 1 mM dithiothreitol).

All proteins were stored in aliquots at −80 °C.

## Co-immunoprecipitation experiments from whole-cell extract

pCSII-EF-MCS plasmids expressing Flag-tagged FANCJ WT or the mutants of interest were transfected into HEK 293T (shown in Figs. 1D, 4A, and 6B) using poly-ethylenimine (PEI, Polyscience, Inc.). At 48 h *post* transfection, cells (about $1 \times 10^9$ cells/experiment) were detached, and collected by centrifugation (250× $g$, 3 min, RT). After two washes in ice-cold PBS, cell pellets were resuspended in Lysis Buffer (50 mM Tris-HCl, pH 8.0, 150 mM NaCl, 0.25% [v:v] Triton X-100, 10% [v:v] glycerol) supplemented with cOmplete EDTA-free protease inhibitor cocktail (Roche). The samples were subjected to sonication on ice (eight cycles consisting of 2-s impulses at an output 15% followed by 5-s intervals), followed by incubation for 20 min at 37 °C in the presence of Benzonase (25 units/sample) and MgCl$_2$

(1 mM). Insoluble material was removed by centrifugation (13000× $g$, 10 min, 4 °C). Then, Flag M2 agarose beads (Merck; volume: 20 μL) were added to cell extract aliquots containing 1–2 mg of total protein. Samples were incubated at 4 °C for 2 h in a rotating wheel. Beads were finally washed four times with Lysis Buffer and pulled-down proteins were eluted by adding SDS-PAGE loading buffer 1.5× (100 mM Tris-HCl, pH 6.8, 20% [v:v] glycerol, 400 mM β-mercapto-ethanol, 1.0% [w:v] SDS, 0.02% [w:v] blue bromophenol) to each pelleted resin sample. Mixtures were incubated at 95 °C for 5 min and subjected to Western blot analysis using the indicated antibodies. Relative Western blot band quantification was performed using the ImageJ software (1.52 v) by analyzing raw grayscale images (8-bit) of three biological replicates of the same experiment performed in the same conditions. The gray mean value of each band and background was measured using the rectangle tool by defining a region of interest that was fitting all the bands. The pixel density for all data (bands and backgrounds) was inverted by subtracting the gray mean value from 255 (pixel value in 8-bit images). The net value (amount) was calculated by deducting the inverted background from the inverted band value. When amount of each band is calculated, AND-1 pulled-down values were normalized to each FANCJ pulled-down values (amount of pulled-down AND-1 divided by the amount of pulled-down FANCJ in each lane). Using this value, to plot the graph, an average of three biological replicates was calculated. The average value was converted to percentage assuming FANCJ WT average value as being 100%.

## In vitro pull-down assays

The pull-down experiments to analyze the interaction of FANCJ with full-length AND-1 (shown in Fig. 2B) were performed in pull-down (PD) Buffer (PBS containing 5% [v:v] glycerol, 0.5 mM TCEP, 0.2% [v:v] Igepal), as previously described (Kilkenny et al, 2017). Saturating quantities of purified His$_{10}$-Myc-tagged AND-1 (12 μg) were added to c-Myc magnetic beads (Pierce™, cat 88842, volume: 20 μL). The beads were washed three times with PD Buffer. Then, purified FANCJ WT or AALA mutant (4 μg) were added. Following a 2-h incubation at 4 °C in the presence of Benzonase (2.40 units/mL; Merck), the beads were washed four times with PD Buffer. Finally, they were resuspended in SDS-PAGE loading buffer supplemented with 2 mM DTT (volume: 30 μL) and boiled for 10 min at 95 °C. Samples were analyzed by western blot using an anti-Myc mouse monoclonal antibody (ab32, Abcam) and HRP-conjugated anti-Flag antibody (A8592, Merck).

Direct interaction between recombinant purified human AND-1 SepB and FANCJ proteins (shown in Figs. 2C and EV1B) was analyzed by co-immunoprecipitation experiments. Flag M2 agarose beads (Merck; volume: 10 μL) were incubated with Flag-tagged FANCJ WT or AALA mutant (2 μg of each purified recombinant protein) in mixtures (final volume: 300 μL) containing pull-down (PD) Buffer 1 (20 mM HEPES-NaOH, pH 7.2, 150 mM NaCl, 5% [v:v] glycerol, 1 mM dithiothreitol, 0.1% [v:v] Igepal) supplemented with cOmplete EDTA-free protease inhibitor cocktail (Roche). Samples were incubated for 1 hr at 4 °C in a rotating wheel. Then, they were washed three times with PD Buffer 1 containing 1% (w:v) BSA and incubated in the same buffer (final volume: 200 μL) for 20 min at 4 °C in a rotating wheel. Purified AND-1 SepB (4 μg) was added to the indicated samples in PD Buffer 1 containing 1% (w:v) BSA supplemented with cOmplete EDTA-free protease inhibitor cocktail (Roche; final volume: 250 μL). Samples were incubated for

1 h at 4 °C in a rotating wheel. Then, samples were washed twice with PD Buffer 1 containing 1% (w:v) BSA and protease inhibitor cocktail and four times with PD Buffer 1. Pulled-down proteins were eluted by adding SDS-PAGE loading buffer 2× (100 mM Tris-HCl, pH 6.8, 20% [v:v] glycerol, 400 mM β-mercapto-ethanol, 1.0% [w:v] SDS, 0.02% [w:v] blue bromophenol) to each pelleted resin sample. Mixtures were incubated at 95 °C for 5 min and subjected to western blot analysis using the indicated antibodies.

GST-pull-down assay to examine the interaction between GST-FANCJ-CIP box and the AND-1 SepB domain (shown in Fig. 2D) were carried out essentially as previously described (Villa et al, 2016). Transformed cells were cultured in LB medium (culture volume: 25 mL) at 37 °C until the A$_{600}$ of the culture reached 0.9–1.0 OD. Expression of the fusion protein was induced by adding IPTG at 0.5 mM to the medium. Cell culture was incubated for 20 h at 16 °C. Cells were centrifuged (3500× $g$, 30 min, 4 °C). The cell pellet was resuspended in Lysis Buffer (50 mM Tris-HCl, pH 7.0, 500 mM NaCl, 10% [v:v] glycerol, 1 mM dithiothreitol) complemented with cOmplete EDTA-free protease inhibitor cocktail (Roche). After cell disruption by sonication, the cell extract was clarified by centrifugation (30000× $g$, 1 h, 4 °C). The soluble extract was then mixed with Glutathione Sepharose 4B beads (50 μL; Cytiva) pre-equilibrated in the same buffer and incubated under rotation at 4 °C for 1 h. Unbound protein was removed by three consecutive washes with Lysis Buffer (1 mL), followed by three washes (1 mL) with PD Buffer 1 containing BSA at 1% [w:v]. Then, the purified AND-1 SepB protein (0.5 mg in 500 μL) was added to the beads and binding was allowed to take place by incubating the mixtures for 1 h at 4 °C under rotation. Thereafter, the beads were washed three times with PD Buffer 1 and washed again three times with the same buffer without BSA. Proteins were eluted by adding SDS-PAGE loading buffer 2× to each pelleted resin sample. Mixtures were incubated at 95 °C for 5 min and subjected to SDS-PAGE followed by Coomassie-staining.

GST-pull-down assays (reported in Figs. 2E and 7A) were carried out to examine the interaction between GST-SepB α-helix bundle and full-length FANCJ WT and mutants of interest. Glutathione Sepharose 4B beads (10 μL) were incubated with GST-SepB α-helix bundle (0.5 μg) in mixtures containing PD Buffer 2 (25 mM Tris-HCl, pH 7.5, 150 mM NaCl and 0.1% [v:v] Triton X-100). Samples were incubated for 1 h at 4 °C in a rotating wheel and washed three times with PD Buffer 2. Then, purified Flag-tagged FANCJ WT and the indicated mutants were added (0.5 μg) to the mixtures containing the beads. Samples were incubated for 1 hr at 4 °C under rotation and finally washed five times with PD Buffer 2 containing NaCl at 300 mM and Triton X-100 at 1% (v:v). Pulled-down proteins were eluted by adding SDS-PAGE loading buffer 2× to each resin pellet. Mixtures were incubated at 95 °C for 5 min and subjected to SDS-PAGE followed by Coomassie-staining or Western blot analysis.

## Generation of a *FANCJ*-knockout (KO) HeLa cell line by CRISPR/Cas9

To establish a FJ-KO line, HeLa cells (ATCCCCL-2) were transiently transfected with SpCas9-expressing pX459 plasmid (Addgene # 62988) using Lipofectamine 2000 (ThermoFisher Scientific). Forty-eight hours *post* transfection, Alt-R synthetic paired guide RNAs (Integrated DNA Technologies), targeting

*FANCJ* exon 2 (TATAAAGCTTACCCGTCACA) and exon 3 (TGTTTGTTGGAGAGTCCCAC), were transfected using Lipofectamine RNAiMAX Transfection Reagent (ThermoFisher Scientific) and expanded for clonal populations. These latter were screened using western blot analysis to detect FANCJ protein expression and further confirmed by Sanger sequencing. For sequencing, *FANCJ* genomic *loci* were amplified using high-fidelity Q5 DNA polymerase (New England Biolabs), and amplified fragments were subcloned using the Zero Blunt Topo PCR cloning kit (Invitrogen). Plasmid DNA was isolated from at least 10–15 colonies and sequenced to identify frameshifts or deletions produced by the Cas9 activity. *CRISPRscan* (https://www.crisprscan.org/) and *CRISPOR* (http://crispor.tefor.net/) online tools were used to design guide RNAs.

## Establishment of stable FJ-KO HeLa cell lines expressing FANCJ WT and the AALA mutant

For establishing FJ-KO HeLa cell lines that express FANCJ WT or the AALA variant, HeLa FJ-KO cells were transfected with the following plasmids: pMK204-TetOne AAVS1-MCS (+) Flag-tagged FANCJ WT or pMK240-TetOne AAVS1-MCS (+) Flag-tagged FANCJ AALA (GenScript). The plasmid pMK240-TetOne AAVS1-MCS (+), was a gift from Masato Kanemaki (National Institute of Genetics, Mishima, Japan) (Okumura et al, 2018). After transfection, cells were cultured in 96-well plates and selected by adding Puromycin (0.3 μg/mL) to DMEM complete medium. Single clones were isolated and further expanded in the presence of Puromycin at a lower concentration (0.2 μg/mL). Expression of Flag-tagged FANCJ, WT and AALA mutant, was assessed before and after induction with Doxycycline (1 μg/mL) by western blot analysis using the indicated antibodies.

All cell lines were tested for mycoplasma contamination and maintained at 37 °C with 5% $CO_2$.

## Cell viability assays

To complement FJ-KO cells, lentiviral particles were produced in HEK 293T by transfecting the pCSII-EF-MCS empty vector (as a control), pCSII-EF-MCS-FANCJ WT and pCSII-EF-MCS-FANCJ AALA individually, together with psPAX2 and pMD2.G. For cell viability assays, HeLa FJ-KO cells, transduced with lentiviral particles, were seeded at a concentration of 1000–1500 cells/well in 96-well plates. Cells were treated with the indicated concentrations of PDS and MMC for 5 days (chronic treatment). Then, after discarding the medium, wells were washed with PBS, and viable cells were detected by adding a crystal violet solution (0.5% [w:v] dissolved in 20% [v:v] methanol) and incubating the plates for 5 min at RT. After removing the crystal violet solution, plates were washed with milli-Q water and dried overnight at RT. Then, absorbance at a wavelength of 570 nm was read in each well using a multi-well plate reader.

## Co-immunoprecipitation experiments from cell nuclear fraction

The association between endogenous FANCJ and AND-1 (as shown in Fig. 1C) was analyzed in HEK 239T cells, cultured in 10-cm dishes. Cells were synchronized in S phase with a single block in thymidine (2 mM) followed by the release in fresh medium for 2.5 h. Preparation of cell nuclear fraction was performed according to a published protocol with modifications (Guillou et al, 2010). Cell pellets were resuspended in 1 mL of Osmotic Buffer (10 mM HEPES-NaOH, pH 7.9, 0.2 M potassium acetate, 0.34 M sucrose, 10% [v:v] glycerol, 1 mM dithiothreitol, 0.1% [v:v] Triton X-100) and incubated for 5 min on ice. After centrifugation (800× g, 5 min, RT), the nucleus/chromatin fraction present in the pellet was resuspended in 1 mL of Hypotonic Buffer (10 mM HEPES-NaOH, pH 7.9, 50 mM NaCl, 1 mM dithiothreitol, 0.1% [v:v] Triton X-100) supplemented with cOmplete EDTA-free protease inhibitor cocktail (Roche). Samples were subjected to sonication on ice (ten cycles consisting of 10-s impulses at an output 10% followed by 20-s intervals) followed by incubation for 20 min at 37 °C in the presence of Benzonase (25 units/sample) and $MgCl_2$ (1 mM). Insoluble material was removed by centrifugation (16000× g, 30 min, 4 °C). Samples (containing 0.5 mg of protein) were used in immunoprecipitation experiments with an anti-AND-1 antibody (H00011169-M01, Novus) bound to Protein A Sepharose beads (Cytiva). Mixtures were incubated for at least 3 hr at 4 °C in a rotating wheel. Beads were washed four times with Washing Buffer (10 mM HEPES-NaOH, pH 8.0, 50 mM NaCl, 1 mM dithiothreitol, 0.1% [v:v] Triton X-100). Proteins bound to the beads were eluted with SDS-PAGE loading buffer, incubated at 95 °C for 5 min, and analyzed by Western blot using the indicated antibodies.

Co-immunoprecipitations (as shown in Fig. 4A) using a CDC45 antibody bound to Protein A Sepharose beads were carried out on nuclear extracts prepared from HEK 293 T cells (about $4 \times 10^7$ cells/ experiment), transiently transfected with the indicated plasmid constructs for FANCJ WT and AALA. After transfection, cell culture synchronization and preparation of cell nuclear fractions was performed as above described.

## Immunofluorescence

For analyzing γ-H2A.X *focus* formation, HeLa cells (*Ctrl*, FJ-KO and complemented cell lines; experiments shown in Fig. 3) were grown overnight on coverslips in six-well plates ($2 \times 10^5$ cells/well), induced with Doxycycline (1 μg/mL) for 24 h and further treated for 24 h with PDS (1 μM) or MMC (10 nM). After incubation, coverslips were fixed in cold methanol for 5 min and further blocked/permeabilized for 1 h at room temperature with PBS containing: 1% [w:v] BSA, 0.3 M glycine, 0.1% [v:v] Tween20. Coverslips were incubated overnight with rabbit monoclonal anti-γ −H2A.X (phospho S139; 1:250, clone EP854(2)Y, Abcam) in a humid chamber at 4 °C. After washing with PBS, coverslips were incubated for 1 h at room temperature with goat anti-rabbit Alexa Fluor™ 488 (1:500, ThermoFisher Scientific) as a secondary antibody. Thereafter, coverslips were mounted into glass slides using mounting media containing DAPI. Images were acquired with a confocal microscope (Zeiss LSM 700) using a ×63 magnification objective (Nikon). γ-H2A.X *foci* were quantified with ImageJ (1.52 v) using the Find Maxima tool with variable values of prominence depending on each experiment: each cell was analyzed, and quantification (number of *foci* identified) made by the software was also checked and confirmed by eye-inspection.

For analyzing γ-H2A.X *focus* formation (experiments shown in Fig. EV3A), HeLa Ctrl and FJ-KO cells were seeded on coverslips in six-well plates ($2 \times 10^5$ cells/well) and grown overnight. The

subsequent day, cells were transfected with the indicated pCSII-EF-MCS plasmids expressing FANCJ WT and its N734H or D736H mutant forms. Transfection was carried out using 1 µg of each plasmid and 3 µL of X-tremeGENE 9 DNA transfection reagent (Roche). In parallel control experiments, carried out in the same conditions but with a pcDNA plasmid expressing eGFP, transfection efficiency was estimated to be not lower than 90%, as revealed by visualizing cells that emitted green light, due to the eGFP presence, versus the total number of cells, observed under a bright field. At 24 h *post* transfection, cells were treated with PDS (1 µM) or MMC (10 nM) for 24 h. After incubation, coverslips were treated, as described before, to detect γ-H2A.X *foci* by indirect immunofluorescence. Thereafter, coverslips were mounted into glass slides using mounting media containing DAPI. Images were acquired with a confocal microscope (Zeiss LSM 980) using a ×63 magnification objective (Nikon) and analyzed with ImageJ (1.52 v), as above described.

## SiRF assays

Quantitative in situ analysis of protein interactions at DNA replication forks (SiRF) assays was carried out as previously described (Roy et al, 2018). HeLa cells (Ctrl, FJ-KO, and stably complemented cell lines) were incubated with Doxycycline (1 µg/mL) for 24 hr before seeding on eight-well chamber slides (Lab-Tek), at ~15,000 cells/well. Cells were pulsed with EdU (125 µM) for 15 min, followed by incubation with hydroxyurea (HU, 4 mM) for stalling DNA replication forks. Cells were subjected to fixation with a solution containing 2% [v:v] paraformaldehyde in PBS for 15 min at room temperature. Then, cells were permeabilized with a solution containing 0.25% [v:v] Triton X-100 in PBS for 15 min at RT, followed by washing with PBS. EdU-biotinylation was performed with a click-it reaction cocktail (PBS containing 2 mM $CuSO_4$, 10 µM biotin-azide, 100 mM sodium ascorbate). Slides were incubated with primary antibodies, mouse anti-biotin (clone BTN.4, ThermoFisher Scientific), and rabbit anti-FANCJ (B1310, Merck). Specific probes that cross-react with the primary antibodies were used for rolling circle amplification and the related SiRF signals were visualized as discrete PLA (proximity ligation assay) spots in cell nuclei that were also counter-stained with DAPI. Images were acquired with a confocal microscope (Zeiss LSM 700) using a 63x magnification objective (Nikon). Signals were quantified using ImageJ software (1.52 v) and the Find Maxima tool with variable values of prominence depending on each experiment. Each cell was analyzed and quantification (number of points identified), done using the software, was also verified by eye inspection. Control experiments were also carried out to detect sites of DNA synthesis in cell nuclei by EdU-EdU PLA spot analysis for the same lines.

## DNA fiber track assays

Cells were pulse-labeled with chloro-deoxyuridine (CldU, 20 µM) for 30 min, followed by incubation in iodo-deoxyuridine (IdU, 200 µM) for 30 min. For the experiments where a PDS treatment was used, the drug was administered at 1 µM to the cells together with IdU. Approximately 1750 cells were lysed with 7.5 µL of Lysis Buffer (200 mM Tris-HCl, pH 7.5, 50 mM EDTA, 0.5% [v:v] SDS). Fibers were spread on Superfrost microscope slides

(Epredia, AG00008332E01MNZ10), which were tilted by ~45° and air-dried for 2 min. Slides were then fixed in a mixture of methanol:acetic acid (3:1, v-v). After drying, or on the next day, DNA was denatured using a solution of HCl (2.5 M) for 1 h. After a careful wash with PBS, slides were incubated with filtered Blocking Solution (PBS containing 1% [w:v] BSA) for 1 h at room temperature. Slides were incubated with a rat anti-BrdU antibody (1:200, clone BU1/75 (ICR1), Abcam) overnight at 4 °C. Next day, slides were washed with PBS and incubated with goat anti-rat antibody labeled with Alexa Fluor™ 647 (1:500, ThermoFisher Scientific). Thereafter, slides were incubated with a mouse anti-BrdU antibody (1:500, clone B44, BD Biosciences), followed by incubation with a goat anti-mouse antibody labeled with Alexa Fluor™ 555 (1:500, ThermoFisher Scientific), both at room temperature for 1 h. Slides were mounted with mounting media containing DAPI (PBS:glycerol, 1:1 [v:v]). Images of DNA fibers were taken with a confocal microscope (Zeiss LSM 700) using a ×63 magnification objective (Nikon). Fiber tract lengths were assessed with ImageJ and µm values were converted into DNA kilobases using the conversion factor 1 µm = 2.59 kb for the spreading technique (Quinet et al, 2017; Nieminuszczy et al, 2016). Sister forks derived from the same fiber were also measured for fork asymmetry analysis.

## DNA helicase assays

PAGE-purified oligonucleotides used for the preparation of DNA substrates were purchased from Merck. An anti-parallel bi-molecular G4 DNA substrate was prepared using the following fluorescent-labeled DNA oligonucleotide OX1 (5'-GACCACTG-[Cy3]T-CGGTTCCAAGCACTGTCGTACTTGATATTTTGGGG TTTTGGGG-3'), as previously described (Calì et al, 2015). DNA helicase assays were carried out in Helicase Assay Buffer (25 mM HEPES-NaOH, pH 7.5, 5 mM MgCl2, 25 mM KCl, 2 mM dithiothreitol, 0.1 mg/mL [w:v] BSA), containing 2 mM ATP and 1.5 nM DNA substrate. Reactions (volume = 20 µL) were initiated by the addition of the indicated proteins. Then, samples were incubated for 20 min at 37 °C. Reactions were quenched with the addition of 5 µL of 5× Stop Solution (0.5% [w:v] SDS, 40 mM EDTA, 0.5 [w:v] mg/mL proteinase K, 20% [v:v] glycerol). Samples were run on a 8% polyacrylamide-bis (29:1) gel in TBE containing 0.1% [w:v] SDS at a constant voltage of 100 V. Both gel and running buffer contained KCl at 10 mM to preserve G4 DNA structure.

For the preparation of the forked duplex DNA substrate, fluorescent-labeled Fluo-D1-35 oligonucleotide (5'-[6FAM] GCA CTGGCCGTCGTTTTACGGTCGTGACTGGGAAAACCCTGGC GTTTTTTTTTTT-3') was annealed to a 3-fold molar excess of the complementary unlabeled D3-35 oligonucleotide (5'-TTTTTTTT TTTTTTTTTTTTTTTTTTTTTTTTTCCCAGTAAAACGACGGCCA GTGC-3'; the complementary sequences are underlined) by incubation at 95 °C for 5 min and gradually cooled to 25 °C. DNA helicase assays were carried out in Helicase Assay Buffer containing 2 mM ATP, 20 nM DNA substrate and 100 nM capture strand oligo, named Cap1 (5'-GCACTGGCCGTCGTTTTAC-3'). Reactions (volume = 20 µL) were initiated by the addition of the indicated proteins and then incubated for 20 min at 37 °C. Samples were quenched with the addition of 5 µL of 5× stop solution. Samples were run on a 8% polyacrylamide-bis (29:1) gel in TBE containing 0.1% [w:v] SDS at a constant voltage of 100 V.

After electrophoresis, gels were analyzed with an imaging system instrument (ChemiDoc, Bio-Rad Laboratories). The displaced oligonucleotide was quantified and any free oligonucleotide in the absence of enzyme was subtracted.

### ATPase assay

The ATPase/GTPase activity assay kit (Sigma-Aldrich, cat. MAK113) was used to measure the presence of inorganic phosphate release from ATP as a measure of FANCJ ATPase activity. The assay mixture (volume: 50 µL) had the following composition: 25 mM HEPES-NaOH pH 7.2, 5 mM $MgCl_2$, 50 mM KCl, 2 mM DTT, 0.1 mg/mL BSA and 5% (v:v) glycerol, 100 nM purified protein, 5 mM ATP with/without 10 nM DNA fork. The DNA fork was prepared using the following oligonucleotides (D1: 5'-CTAC TACCCCCACCCTCACAACCTTTTTTTTTTTTTTT-3'; D2: 5'-TT TTTTTTTTTTTTTGGTTGTGAGGGTGGGGGTA-3'; the complementary sequences are underlined). Samples were incubated at 37 °C for 30 min. After incubation, Malachite Green Reagent solution (volume: 200 µL) was added to each sample and incubation continued at room temperature for additional 10 min. $A_{635}$ was measured using the Infinite F200 PRO TECAN instrument. Concentrations of inorganic phosphate were determined using a standard curve derived from known phosphate concentrations.

### Statistical analysis

Statistical significance (*P* values), analysis, and graphs were performed using GraphPad Prism 6 software. Statistical tests and differences are mentioned in the respective figure legend.

### Antibodies

The following commercial primary antibodies were used: anti-Flag peroxidase-conjugated mouse monoclonal antibody (Sigma-Aldrich cat. A8592); anti-AND-1 mouse monoclonal antibody (Novus cat. H00011169-M01); anti-FANCJ rabbit polyclonal antibody (for Western blot analysis; Novus cat. NBP1-31883); anti-Myc mouse monoclonal antibody (ab32, Abcam); anti-FANCJ rabbit polyclonal antibody (for SiRF analyses; Sigma-Aldrich cat. B1310); anti-poly-Histidine peroxidase-conjugated mouse monoclonal antibody (Sigma-Aldrich cat. A7058); anti-GST antibody (Cytiva cat. RPN1236); anti-β-tubulin mouse monoclonal antibody (Elabscience cat. E-AB-20033); anti-γ-H2AX rabbit monoclonal antibody (Abcam cat. ab81299); anti-biotin mouse monoclonal antibody (ThermoFischer Scientific cat. MA5-11251); anti-MCM4 rabbit polyclonal antibody (Abcam cat. Ab4459); anti-BrdU rat monoclonal antibody (Abcam cat. ab6326); anti-BrdU mouse monoclonal antibody (BD Biosciences cat. 347580); anti-BRCA1 rabbit polyclonal antibody (Proteintech cat. 22362-1-AP); anti-BLM mouse monoclonal antibody (Santa Cruz Biotechnology cat. sc-365753); anti-MLH1 rabbit monoclonal antibody (Bethyl laboratories cat. A700-112-T).

The following secondary antibodies were used: anti-mouse IgG peroxidase-conjugated goat polyclonal antibody (ThermoFisher Scientific cat. 31439); anti-rabbit IgG peroxidase-conjugated goat polyclonal antibody (ThermoFisher Scientific cat. 31462); anti-rabbit goat antibody labeled with Alexa Fluor™ 488 (ThermoFisher Scientific cat. A11008); anti-rat goat antibody labeled with Alexa Fluor™ 647 (ThermoFisher Scientific cat. A21247); anti-mouse polyclonal goat antibody labeled with Alexa Fluor™ 555 (Thermo-Fisher Scientific cat. A21422).

## Data availability

This study includes no data deposited in external repositories. All the materials described are available upon request.

## Peer review information

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

## Acknowledgements

The authors acknowledge Hisao Masai (Tokyo Metropolitan Institute of Medical Science, Tokyo, Japan) for the pCSII-EF-MCS plasmid expressing Flag-tagged FANCJ WT; Masato Kanemaki (National Institute of Genetics, Mishima, Japan) for the plasmid pMK240-TetOne AAVS1-MCS (+); Luca Pellegrini (University of Cambridge, Cambridge, United Kingdom) for the plasmid pRSF-Duet-1 expressing 6xHis-SepB and pFBDM for producing full-length 10xHis-Myc-tagged AND-1. Alessandro Vindigni (Washington University in Saint Louis, USA) and Roberto Bellelli (Queen Mary University of London, United Kingdom) are acknowledged for helpful discussions. This study has received funding from the European Union's Horizon 2020 research and innovation program under Marie Skłodowska-Curie Actions [AntiHelix - Grant Agreement n. 859853 to AB, DS, FMP, LMRN, and SO; Syntrain - Grant Agreement n. 722729 to NKJ and DB); from the Cross-border Cooperation Programme Italy-Slovenia 2014–2021 by the European Regional Development Fund and national funds (TransGlioma), and from *Associazione Italiana per la Ricerca sul Cancro* [AIRC IG n. 25965 to FMP; AIRC IG 23710 and IG 18976 to DB; AIRC IG 20778 to SO].

## Author contributions

**Ana Boavida**: Data curation; Formal analysis; Investigation; Writing—original draft; Writing—review and editing. **Luisa MR Napolitano**: Data curation; Investigation; Methodology; 2B,D and EV1B. **Diana Santos**: Data curation; Investigation; Methodology; DS purified the AND-1 6xHis-SepB. **Giuseppe Cortone**: Data curation; Investigation; Methodology; GC produced the pCSII-EF-MCS plasmid expressing Flag-tagged FANCJ AALA mutant, carried out some of the co-immunoprecipitations in cell extracts and the DNA damage analysis shown in Fig. EV4B. **Nanda K Jegadesan**: Data curation; Investigation; Methodology; NKJ generated the FJ-KO HeLa cell line and carried out cell survival assays. **Silvia Onesti**: Supervision; Funding acquisition; Writing—original draft; SO produced the structural models. **Dana Branzei**: Supervision; Funding acquisition. **Francesca M Pisani**: Conceptualization; Data curation; Supervision; Funding acquisition; Writing—original draft; Project administration; Writing—review and editing.

## Disclosure and competing interests statement

The authors declare no competing interests.

# Expanded View Figures

**A**

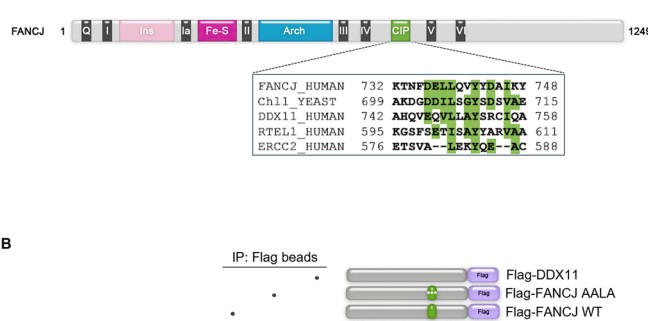

**B**

**Figure EV1.  FANCJ, but not DDX11, directly interacts with AND-1 6xHis-SepB.**

(**A**) Schematic representation of the polypeptide chain of human FANCJ. Conserved sequence motifs and domains are indicated with the same abbreviations and colors used in Fig. 1A. In the insert, alignment of human FANCJ and budding yeast Chl1 CIP box sequence is reported with corresponding regions of other SF2 Fe–S human DNA helicases. Abbreviations used are *HUMAN, Homo sapiens* and *YEAST, Saccharomyces cerevisiae.* The KALIGN tool (version 3.3.1) was used. Highly conserved amino-acid residues are highlighted in green. (**B**) Co-pull-down experiments of Flag-tagged FANCJ WT or AALA mutant and AND-1 6xHis-SepB using anti-Flag agarose beads. Pulled-down samples were analyzed by Western blot with an anti-Flag peroxidase-conjugated mouse monoclonal antibody (Sigma-Aldrich cat. A8592) and anti-poly-Histidine peroxidase-conjugated mouse monoclonal antibody (Sigma-Aldrich cat. A7058). Schematic of the recombinant proteins used in the pulled-down experiments is shown.

**A**

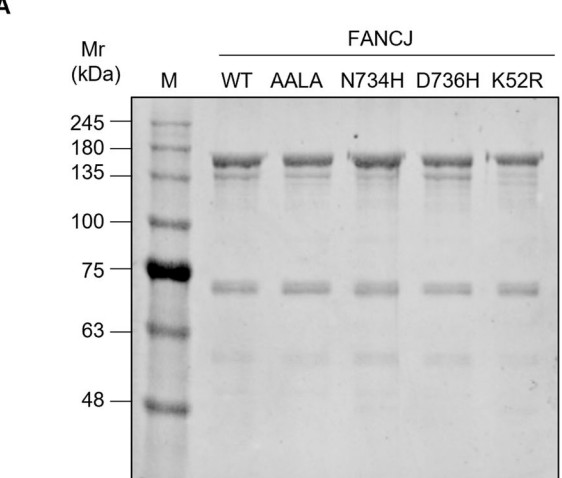

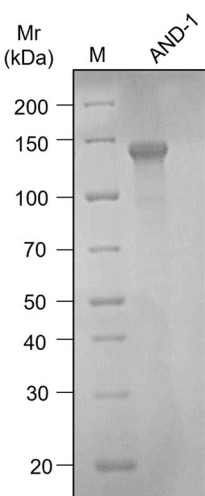

**B**

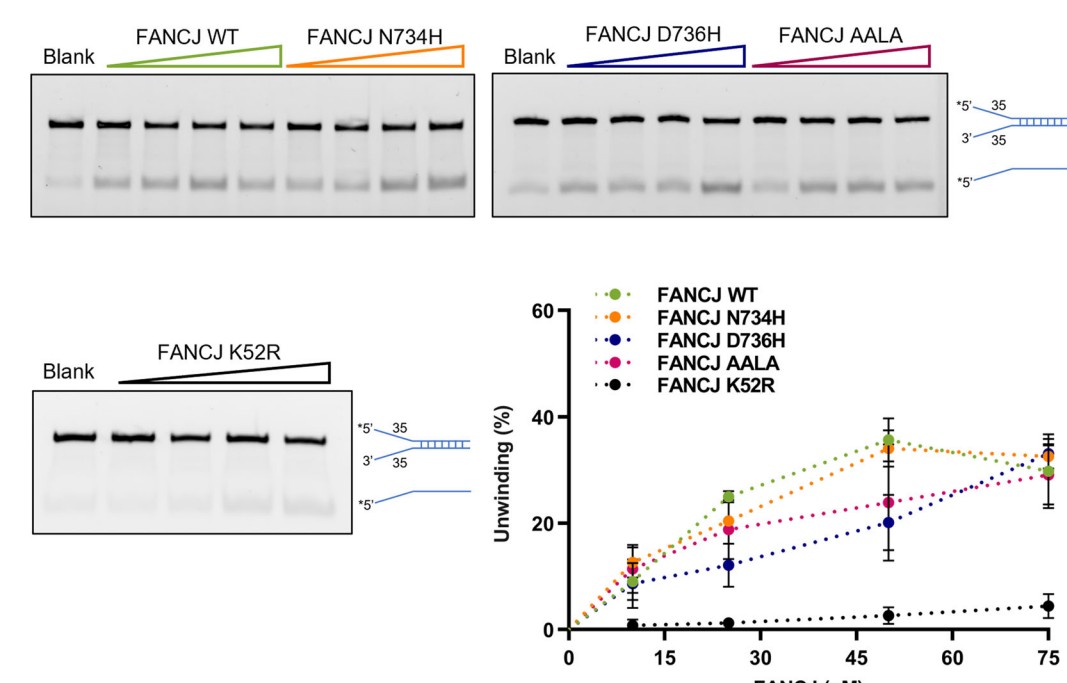

**C**

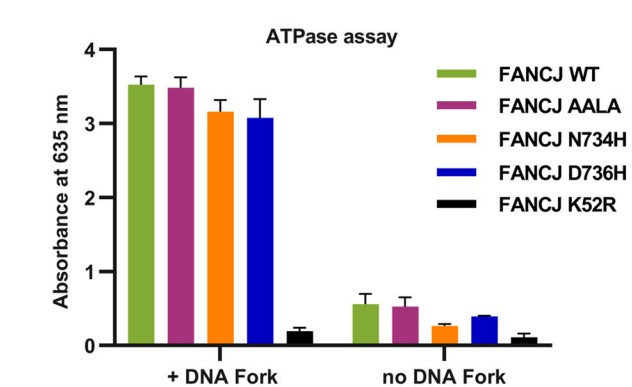

◀ **Figure EV2.   Biochemical analysis of CIP box FANCJ mutant derivatives.**

(A) SDS-PAGE analysis of purified recombinant FANCJ WT and indicated mutants and AND-1 full-length protein. An aliquot of each protein batch (3 μg) was loaded onto the indicated gel lane. Gels were stained with ProBlue Safe Stain (GiottoBiotech). Size of protein markers, loaded onto the lane indicated with *M*, is reported on the left. (B) DNA helicase assays using a forked duplex fluorescent-labeled DNA substrate. Asterisk represents the fluorophore attached to the 5′-end of the DNA oligonucleotide, named Fluo-D1-35. Blank refers to a control assay without protein. Data plot derives from three independent experiments (mean ± SD). (C) ATPase assays were carried out using the indicated FANCJ proteins (100 nM) with/without the DNA fork ligand, as described in "Methods". Values reported derive from three independent experiments (mean ± SD).

**A**

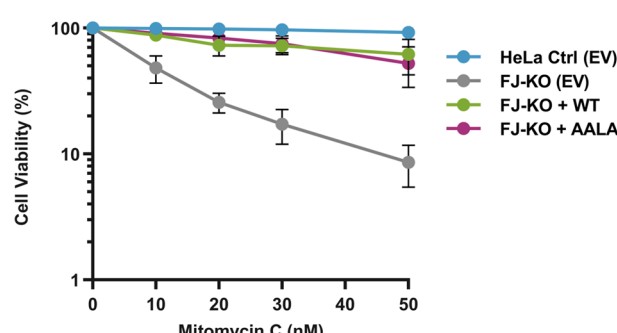

**B**

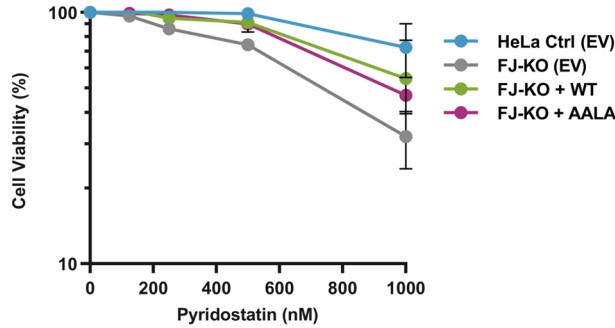

**C**

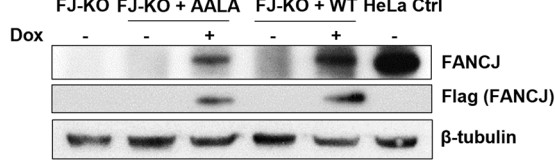

**Figure EV3. Establishment of a FJ-KO HeLa cell line and its complementation by *FANCJ* WT and AALA mutant alleles.**

(A) Schematic representation of the human *FANCJ* genomic *locus*. The sequence of CRISPR-paired guide RNAs targeting exons 2 and 3 is reported. *PAM* sequence is highlighted in red. (B) Viability assays of FJ-KO HeLa cell lines complemented with FANCJ WT and the AALA mutant. HeLa cells (*Ctrl*) or FJ-KO cells, transduced with lentiviral particles deriving from plasmid pCSII-EF-MCS-FANCJ WT or -FANCJ AALA or the empty vector (*EV*), were treated for 5 days with the indicated concentrations of MMC and PDS ($n = 3$ biologically independent experiments, mean ± SD). Cells were detected by crystal violet staining, as described in "Methods". Paired *t* tests were performed to analyze statistically significant differences between the HeLa control and the complemented cell lines, but no differences were found for both viability assays. (C) FJ-KO HeLa cell lines were established that stably express Flag-tagged FANCJ WT or the AALA mutant under the control of a Tetracycline-responsive promoter. Expression of ectopic *FANCJ* was detected before and after induction with Doxycycline (1 μg/mL; *Dox*) by Western blot analysis of whole extracts from the indicated cell lines using an anti-FANCJ or anti-Flag antibody. β-tubulin was used as a loading control.

A

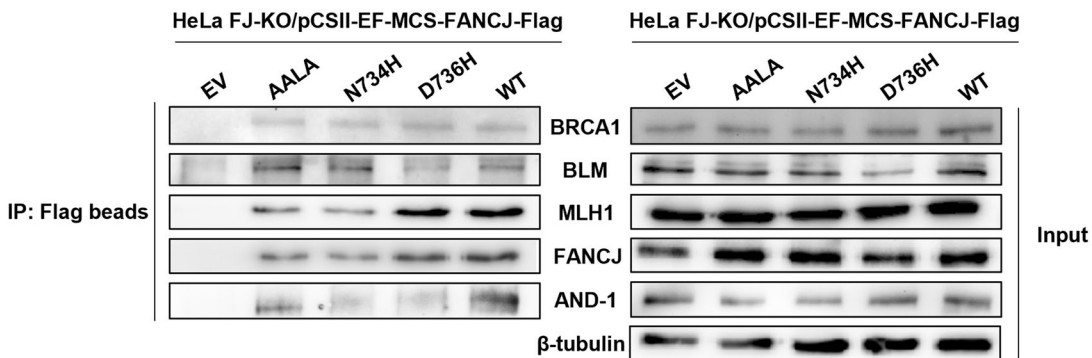

B

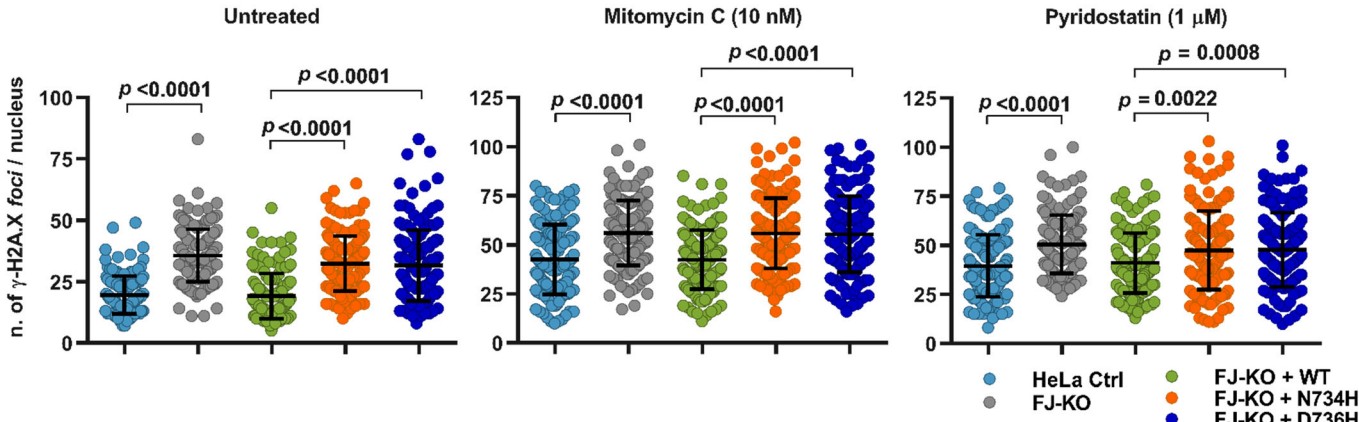

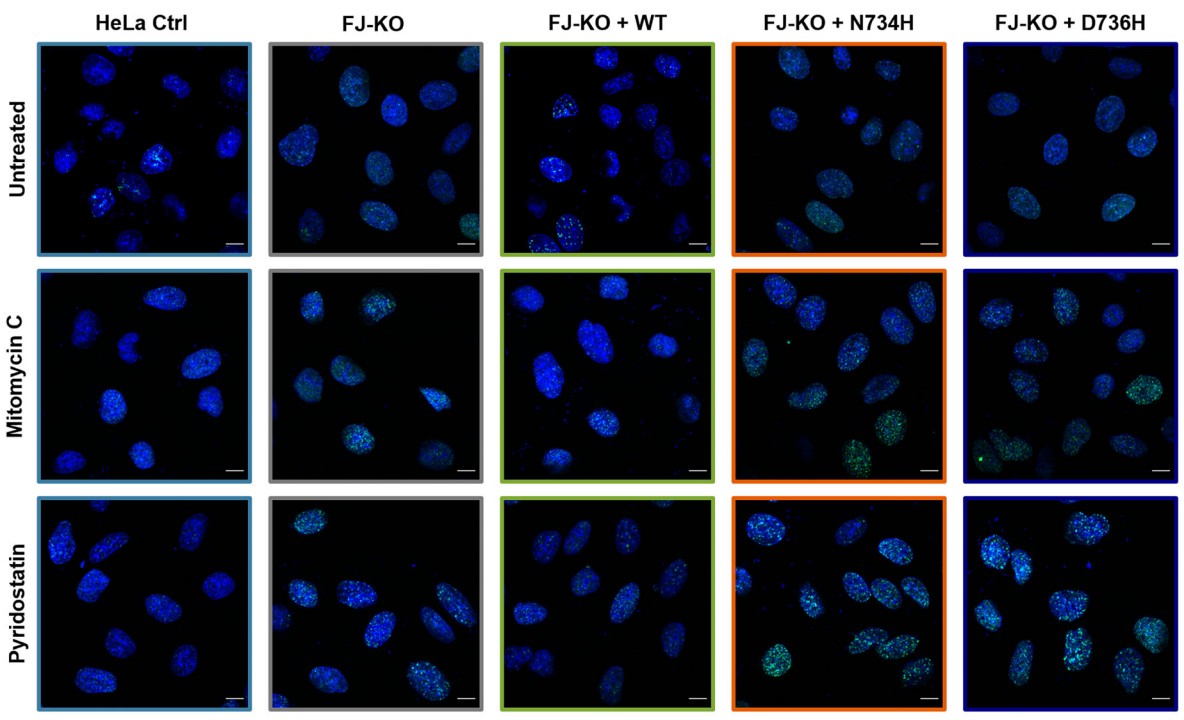

◀ **Figure EV4.  FANCJ CIP box mutants, AALA, N734H and D736H retain the ability to interact with BRCA1, BLM and MLH1 and induce enhanced DNA damage.**

(A) Co-immunoprecipitation experiments with anti-Flag agarose beads were carried out on whole extracts of HeLa FJ-KO cells transiently transfected with pCSII-EF-MCS vector expressing Flag-tagged FANCJ wild type (*WT*) or the indicated mutants (*EV* stands for empty vector). Western blot analysis was carried out on the input (0.6% of each sample; 10 μg of total protein) and pulled-down material (50% of each sample). Proteins of interest were detected by Western blot experiments using the indicated antibodies. Experiments were done in triplicate. (B) HeLa FJ-KO cells, transfected with plasmid vectors expressing FANCJ WT or the indicated mutants, were treated with PDS or MMC. γ-H2A.X *focus* formation was detected by immunofluorescence with a monoclonal antibody that specifically recognizes the Ser139-phosphorylated form of the above histone. Scale bar, 10 μm. Dot plot of the number of *foci* detected per cell is reported in each graph. Bars indicate mean ± SD; 150 cells were analyzed per condition; $n = 2$ biologically independent experiments, with at least two technical replicates per experiment; two-tailed *P* value ($P < 0.01$) was calculated using Student's t test nonparametric for unpaired data with Welch correction.

