## [Peer Review File · EMBO Reports]

FANCI DNA helicase is recruited to the replisome by AND-1 to ensure genome stability

Ana Boavida, Luisa Napolitano, Diana Santos, Giuseppe Cortone, Nanda Jegadesan, Silvia Onesti, Dana Branzei, and Francesca Pisani

DOI: [10.15252/embr.202358282](https://doi.org/10.15252/embr.202358282)

Corresponding author(s): *Francesca Pisani (francesca.pisani@ibbc.cnr.it)*

Review Timeline:

Submission Date:	9th Oct 23
Editorial Decision:	16th Oct 23
Appeal Received:	17th Oct 23
Editor's Correspondence:	17th Oct 23
Author's Correspondence:	19th Oct 23
Editorial Decision:	22nd Nov 23
Revision Received:	5th Dec 23
Accepted:	15th Dec 23

Editor: *Esther Schnapp*

Transaction Report:

Dear Dr. Pisani,

Thank you for the submission of your manuscript to EMBO reports. I have now read and discussed it with my colleagues here, including our chief editor Bernd Pulverer, and I am sorry to say that we all agree that the ms is not a good fit for us.

We note that your study reports that FANCI directly interacts with AND-1 through a highly conserved CIP box, and that this interaction is required for preserving genome stability and counteracting DNA replication stress in human cells. You also show that cancer-associated CIP box variants of FANCI show reduced binding to AND1. We think that this will be of interest to researchers in the field. However, we also note that it was reported before that the *S. cerevisiae* Chl1 DNA helicase is recruited to DNA replication forks through a direct interaction with Ctf4/AND-1, as you mention. It further remains unknown whether the cancer-associated FANCI CIP box mutants play a causative role in cancer formation.

We think that the manuscript does not provide a sufficient advance and is not sufficiently developed for consideration for publication here, and we have therefore decided not to proceed with in-depth review.

Having said that, we hope you consider our not-for-profit open access sister journal, Life Science Alliance (LSA), as a venue for your work (<https://www.life-science-alliance.org/about-journal>). We have not discussed the paper with the LSA editorial team, but we believe that your work fits the scope of the journal and encourage you to transfer your manuscript to LSA using the link below. Should you transfer, LSA editors will send your manuscript out for peer-review. All files and manuscript details will be automatically transferred, and you will have the opportunity to include specific comments or requests for the LSA editors before finalizing transfer. Please copy our decision letter text in the comments to the LSA editors, so that they are aware that we have committed peer-review on behalf of LSA for this manuscript. Feel free to reach out to the LSA executive editor, Eric Sawey (e.sawey@life-science-alliance.org) if you have any questions about the LSA journal or the transfer process.

For EMBO reports, I am sorry that I cannot be more positive this time, and I thank you once more for your interest in our journal.

Yours sincerely,

** As a service to authors, EMBO Press provides authors with the ability to transfer a manuscript that one journal cannot offer to publish to another journal, without the author having to upload the manuscript data again. To transfer your manuscript to another EMBO Press journal using this service, please click on
Link Not Available

Dear Dr Esther Schnapp,

thanks for sending the email with the EMBO reports editorial decision. I would like to express my disappointment and frustration about this rejection that is really unexpected by all the authors.

I would like to make the following points.

One of the reasons you have rejected the manuscript is the lack of novelty of our findings, as you have written that

"it was reported before that the S. cerevisiae Chl1 DNA helicase is recruited to DNA replication forks through a direct interaction with Ctf4/AND-1".

It should be pointed out that the human ortholog of the yeast Chl1 DNA helicase is DDX11 and not FANCD1. In our manuscript we demonstrate that FANCD1, and not DDX11 (see Figure EV1), directly interacts with AND-1, the human counterpart of yeast Ctf4 (and *via* this interaction it is recruited to the DNA replication machinery). This finding is by itself novel and important, compared to what has been reported for the budding yeast system. Both DDX11 and FANCD1 are important DNA helicases that play key roles in DNA replication/repair pathways and are linked to genome instability hereditary disorders (Warsaw breakage syndrome and Fanconi anaemia, respectively). Besides, FANCD1 is the third most mutated gene in hereditary breast and ovarian cancers and our biochemical study of some FANCD1 cancer-related missense variants (previously considered VUS, variants of unknown significance) could be of great interest for molecular oncologists and in general for physicians. I would like to point out that it is out of the scope of our present work testing if the FANCD1 mutants of interest are causative of cancer (more likely they are risk factors for some kind of tumors), as this would require the creation and analysis of appropriate murine models. Nonetheless, our results offer a possible interpretation of the functional significance of the cancer-relevant FANCD1 variants, which have been uncharacterized so far.

An additional important consideration is that a previous version of this same work was submitted to EMBO journal and sent to reviewers by Hartmut Vodermaier, Senior Editor of EMBO journal (in cc). It is now really disappointing that an improved version of the same work is desk rejected by EMBO reports.

All that considered I would like to ask you to reconsider your decision about our manuscript. We believe our work deserves to be evaluated by expert referees in the field and I wonder whether you are willing to consult Karim Labib, member of the EMBO reports Advisory Editorial Board, about the suitability of our manuscript for the EMBO reports standard in terms of novelty and interest in the DNA replication/genome stability field, even considering that he published a paper on yeast Ctf4-binding proteins some years ago [Villa F, Simon AC, Ortiz Bazan MA, Kilkenny ML, Wirthensohn D, Wightman M, Matak-Vinković D, Pellegrini L, Labib K. Ctf4 Is a Hub in the Eukaryotic Replisome that Links Multiple CIP-Box Proteins to the CMG Helicase. Mol Cell. 2016 Aug 4;63(3):385-96].

I look forward to hearing from you about

Best regards

Francesca Pisani

Francesca M. Pisani

Research Director

IBBC - CNR

Via P. Castellino, 111

80131 - Naples. Italy

Tel: +39-0816132292 (Office)

+39-0816132246 (Laboratory)

+39-3396313841 (Mobile)

Coordinator of the H2020-MSCA ETN AntiHelix

Coordinator of the HE-MSCA DN CohesiNet

From: Esther Schnapp | EMBO Reports <e.schnapp@emboreports.org>

Sent: 17 October, 2023 2:53 PM

To: Francesca M. Pisani <francesca.pisani@ibbc.cnr.it>

Subject: Re: Decision on Manuscript EMBOR-2023-58282V1

Dear Dr. Pisani,

Thank you for your email asking us to reconsider our decision on your ms.

I have re-discussed your study with my colleagues now, including our chief editor Bernd Pulverer. Given that your ms was already peer-reviewed at EMBO press and thus has been evaluated by experts in the field, we recommend to handle it as a transfer, which is also in line with our policy to save referees time.

If you agree, can you please send us a point-by-point response to the existing referee reports?

Thank you and best regards,

Esther

Point-by-point response to all the issues raised by the *EMBO Journal* Referees

Referee #1:

The paper by Boavida et al reports a novel interaction between the DNA repair helicase FANCI and the replisome component AND-1. The authors identify a putative CIP-box sequence in FANCI and provide biochemical evidence that it mediates the interaction with the Sep1 domain of AND-1. They further show cellular evidence that FANCI is recruited to the replisome in an AND-1 dependent way and that the FANCI-AND-1 interaction is important for coping with replicative stress. The topic of how the DNA synthesis machinery copes with obstacles to replication such as transient secondary structure formation in the DNA, which are known to be dealt with by enzymes such as FANCI, is the current subject of intense research efforts, so the evidence reported in this paper is timely and of interest to the field.

The experiments appear to have been well executed and the data are of good quality. The evidence seems all consistent with the proposed interaction and its role in replisome function. I have one, significant concern: the putative CIP sequence identified by the authors is part of a folded domain in FANCI and the hydrophobic residues that would be expected to interact with AND-1 are buried (see picture attached). In order to interact with AND-1, the CIP helix of FANCI would need to rearrange its conformation in a major way. This is not impossible, but it's not what has been seen before, where the CIP sequence is normally unstructured and exposed. If the paper is to be published, it is important that the authors address this point in the Discussion: they should mention the possibility that FANCI CIP might just happen to be part of AND-1 binding surface, and that the interaction might actually be different from the mode of CIP binding observed before. This issue will be resolved by the structure of the FANCI-AND-1 complex, which is beyond the scope of the present study.

We thank Referee #1 for the issues raised regarding the structural features of the FANCI CIP box we have described in our work. Our reply to his/her criticism is reported below.

Referee #1 points out that the α -helix that includes the putative CIP box is part of the helicase core, and that some of the residues expected to be involved in the canonical mode of interaction between with AND-1/Ctf4 are partially buried both in the FANCI and Chl1 AlphaFold models.

Indeed, we have clearly stated that this is the case, in the Discussion section (pages 16-17 of the manuscript), and we have suggested a number of possible alternatives to account for this, including:

- (a) the possibility that the AF models are not fully accurate
- (b) the possibility that the α -helix (that is partially exposed to solvent and is not an integral part of the hydrophobic core) may undergo a conformational change
- (c) the possibility that the interaction with AND-1 does not involve the canonical framework observed with CIP boxes that are located in low complexity sequences, but rather the external face of the α -helix.

While we agree that the first option is probably weak, and we have not mentioned it in the manuscript, we think both the other two alternatives are possible. Regarding option (b), structural biology has shown, multiple times, that protein structures are not rigid, and that much larger conformational changes are possible, including the remarkable reorganization of secondary structure elements. As for option (c), this is compatible with our mutagenesis results showing that the exposed residues N734 and D736 are critical for binding.

We could add the following Expanded View Figure to a revised version of our manuscript, where a structural model of the FANCI CIP box α -helix is shown. Side chains of highly conserved amino-acid residues are shown as sticks.

In our manuscript, we present strong *in vivo* and *in vitro* data to show that there is a physical and functional interaction between FANCD1 and AND1. In particular, we show that mutations in the FANCD1 CIP-box α -helix affect AND1-binding without disrupting the DNA helicase activity: the latter point provides strong evidence that the overall fold of the protein is not affected, and that the effect of the mutations does not simply disrupt the protein.

We wish to stress that the same situation exactly occurs for the Chl1/Ctf4 interaction, where the putative CIP-box α -helix is expected to be in the same conformation as the FANCD1 CIP-box. As in our manuscript, this interaction has been demonstrated by *in vitro* and *in vivo* data (Samora *et al.*, Molecular Cell, 2016), and has been validated by subsequent work (including Villa *et al.*, Molecular Cell, 2016). A picture with the α -helix, clearly leaning against the Chl1 helicase core, is indeed found in the graphical abstract of Samora *et al.* and both papers include among the authors some of the most competent and well recognized experts in the structural biology of DNA helicases. We are therefore not alone in finding that the evidence for a CIP-box in some FeS helicases is plausible and compatible with the structural data.

In the helicase assays of Figure 7D, the authors should show a control experiment with ATP, to show that G4 DNA unfolding requires helicase activity and not just DNA binding. This is to rule out the possibility that FANCD1 binding alone might separate the two oligos that make up the G4. It's not clear in the figure legend what the 'blank' lane represents.

The DNA helicase control reaction without ATP is now present in Figure 7B (last lane on the right in the gel for FANCD1 wild type protein) and shows no G4 DNA unfolding in the absence of ATP.

Explanation of the "Blank" lane is now given in the Figure legend.

 Referee #2:

This manuscript entitled 'FANCD1 helicase is recruited to the replisome by AND1 to ensure genome stability' by Boavida *et al.* describes a novel interaction between FANCD1 and the replisome component AND1. The data presented indicate that FANCD1 associates with the replication fork through this interaction. Expression of a FANCD1 mutant that disrupts this interaction enhances the number of gammaH2AX foci, both in untreated cells and cells treated with MMC or Pyridostatin. Therefore, the authors conclude that this interaction is required to prevent DNA damage. Loss of this interaction also reduces replication fork speed and causes fork asymmetry. Finally, a number of FANCD1 variants found in cancer cells were shown to affect AND1 binding suggesting that loss of the interaction could be involved in cancer predisposition.

FANCD1 is an important helicase that is linked to several pathologies including cancer and Fanconi anemia. It is known to act in ICL repair, homologous recombination, and the resolution of alternative DNA structures but its exact role in several of these processes is not completely understood. There are a number of indications in the literature that FANCD1 is present at the replication fork but how it is recruited there and whether this presence is constitutive or damage-induced is not clear. The identification of the FANCD1-AND1 interaction and demonstration that this interaction is important for the localization of FANCD1 to the replisome is important to further our understanding of the function of this helicase. The data provided to demonstrate this interaction is solid and there are a number of indications presented that link this interaction with genome stability. While I would definitely consider this manuscript for publication in EMBO journal, there are several major and some minor issues that would need to be resolved first.

We thank Referee #2 for highlighting our findings hold value and could be significant in the DNA replication/genome stability field.

1) Several FANCD1 mutants are produced by expression of the Flag-tagged proteins in mammalian HEK293 cells and isolation of the proteins on flag-resin. Helicase activity was measured in gel-based assays and the results were quantified. Based on the results presented in Figure 7 and Figure S3 the authors conclude that the helicase activity of the mutants is very similar to the WT. However, based on the gels and the quantification graphs there does seem to be a considerable difference in activity between the proteins (the values at some protein concentrations vary 2-fold or even more). Although I realize it is difficult to extract the exact catalytic activity from these graphs, the authors should at least come up with a way to add a numeric value to the apparent catalytic activity to make this convincing. Furthermore, the relative difference is difficult to judge without a mutant protein that does affect catalytic activity. Therefore, the authors should at least include a catalytic dead mutant (also to make sure the activity they observe is not due to a co-purified protein from the cell extract) and preferably also a mutant that is partially affected. It should be discussed why the differences they observe are considered to be small.

We have produced the FANCD1 K52R ATPase/helicase-dead mutant and measured its DNA helicase (Figure 7B and EV2A-B) and ATPase (Figure EV2C) activity in parallel with the other FANCD1 mutants of interest. As described in the *Results* section (page 8), FANCD1 K52R “was found to have a very low level of DNA unwinding activity, which could be ascribed to a residual catalytic function observed only at high protein concentrations. Moreover, FANCD1 K52R ATPase activity was almost completely abolished and not stimulated by DNA, as observed for the ATP hydrolysis catalyzed by FANCD1 WT and AALA mutant in side-by-side experiments (Figure EV2C). Therefore, these results have allowed us to ascribe the observed DNA unwinding activity to the purified recombinant FANCD1 WT or the mutant proteins and not to any co-purified contaminant DNA helicase.”

DNA unwinding activity of the FANCD1 mutants of interest is reported in the plots of Figure 7B and EV2B and described in the *Results* section (page 8 and page 15-16).

2) While the authors show more gammaH2AX foci in cells expressing a FANCD1 mutant that does not interact with AND-1, it is not shown how this affects the cells. If this results in enhance genome instability you would expect to see an effect on cell viability or chromosomal integrity but these parameters are not shown. To conclude that this interaction is important to ensure genome stability the authors should show the viability assays as shown in Figure 3 with the WT and mutant (AALA) proteins expressed in the FANCD1-KO cell line. Furthermore, it would strengthen their claim if they could show a direct effect on chromosomal integrity (by chromosomal spread assays) in FANCD1 KO cell lines expressing the wt and mutant protein. To determine whether this potential effect on chromosomal integrity is independent on one of the canonical functions of FANCD1 in HR, the authors should test the activity of this mutant in the DR-GPF assay.

We have addressed this important point raised by Referee #2 as described below.

Instead of performing cell viability assays, we have analyzed proliferation curves of the various complemented cell lines after treatment with either MMC or PDS by cell culture time-lapse imaging using an Incucyte® system (Sartorius). What emerges from this analysis is that the FJ-KO line has a clear proliferation defect that is further exacerbated after treatment with MMC and PDS. This phenotype is partially reverted by expressing FANCD1 WT but not the AALA mutant derivative. The Figure below can be added into a revised version of our manuscript in the EV data.

Expanded View Figure. FANCI loss or reduced binding to AND-1 lower cell proliferation. The indicated cell lines were cultured in 96-wells plates and treated with MMC (10 nM) or PDS (1 μM) for 2 hr. Then, after a washout with PBS to remove the drugs, fresh medium containing Doxycycline was added to each well and replaced after 48 hr. Growth was monitored using an Incucyte® system (Sartorius) in three technical replicates (mean ± SD).

To address the issue of whether involvement of the FANCI CIP box mutants of interest in other DNA repair/recombination pathways is affected, we have analyzed if they retain the ability to interact with BRCA1, BLM and MLH1 by co-IP experiments in cell extracts. Our results have revealed that the tested FANCI mutants still associate with all the above known interacting partners, suggesting that the relevant functions could be not altered by the FANCI CIP box mutations. The Figure below can be added into a revised version of our manuscript in the EV data.

Expanded View Figure. CIP-box mutations do not abolish binding of FANCI to BRCA1, BLM and MLH1. Co-immunoprecipitation experiments with anti-Flag agarose beads were carried out on whole extracts of HeLa FJ-KO cells transiently transfected with pCSII-EF-MCS vector expressing Flag-tagged FANCI wild type (WT) or the indicated mutants (EV stands for empty vector). Western blot analysis was carried out on the input (0.6% of each sample; 10 mg of total protein) and pulled down material (50% of each sample).

3) An important issue that is not clear from the manuscript is whether this FANCD1-AND1 interaction is constitutive or damage-induced. Based on the data presented in Figure 5 this interaction already exists without adding external damage to the cells. However, it could be that this interaction is mediated by endogenous damage/secondary DNA structures. Therefore, it is important to show what happens to this interaction upon induction of damage. If this enhances this interaction it might point to a damage-dependent mechanism that might be explained by some of the pathways FANCD1 is known to act in. If it is not enhanced by damage it indicates this interaction is constitutive and points to a more general role for FANCD1 in proper DNA replication which would be interesting and novel. Therefore, including a DNA damage condition (MMC and pyridostatin) to the PLA assay in Figure 5B is important.

This experiment has not been carried out.

4) The FANCD1(ALAA) mutant seems to have a dominant effect. Its expression enhances the gammaH2AX foci and the reduction in replication fork speed. This is difficult to understand since lack of interaction should not give an enhanced defect compared to a knockout. A possible explanation of this observation should at least be discussed.

We agree with Referee # 2 that the dominant-negative behavior of the FANCD1 ALAA mutant should be discussed. One possible explanation of the observed phenotype is that alternative DNA damage repair pathways are not activated, when the FANCD1 DNA helicase is present in cell nuclei, although not directly associated to the replication machinery. This interpretation can be added in a revised version of the present manuscript.

5) In Figure 7, several FANCD1 CIP-box mutants are described that were found in cancer cells. Some of these mutants also show reduced binding to AND-1 (see also my last minor point below). Since they showed in previous figures that preventing this interaction induces DNA damage signalling and therefore possibly genome instability, it is concluded that clinically relevant FANCD1 CIP box mutations could be cancer risk factors. However, no functional experiments, such as gammaH2AX foci formation or replication fork speed, were shown for any of these mutants. Therefore, this conclusion is based on a single mutant (ALAA) that disrupts this interaction. The claim that reduction of the FANCD1-AND1 interaction induces a functional defect that could lead to genome instability would be much stronger if this is also shown for the cancer mutants.

We have evaluated the effect of the cancer-related FANCD1 N734H or D736H variants in DNA damage induction. We have examined γ -H2A.X focus formation in FANCD1-KO lines complemented with FANCD1 WT or the above FANCD1 mutants, with/without treatment with PDS or MMC. In a revised version of our manuscript, we could add the Figure below showing that enhanced DNA damage is counteracted only in the FJ-KO line complemented with FANCD1 WT, but not with the FANCD1 N734H or D736H mutants.

Expanded View Figure. FANCI CIP box mutants N734H and D736H induce enhanced DNA damage. HeLa FJ-KO cells, transfected with plasmid vectors expressing FANCI WT or the indicated mutants, were treated with PDS or MMC. γ -H2A.X focus formation was detected by immunofluorescence with a monoclonal antibody that specifically recognizes the Ser139-phosphorylated form of the above histone. Scale bar, 10 μ m. Dot plot of the number of foci detected per cell is reported in each graph. Bars indicate mean \pm SD; 150 cells were analyzed per condition; $n = 2$ biologically independent experiments, with at least 2 technical replicates per experiment; two-tailed p -value ($p < 0.01$) was calculated using Student's t -test non-parametric for unpaired data with Welch correction.

6) All cell biological experiments are performed in HeLa cells. It is important to confirm these results with another cell type.

Although we acknowledge the importance of doing the experiments in another cell type, this is usually not a common practice for most of the experiments we have performed. Furthermore, considering the establishment of stable cell lines, this would be extremely time-consuming.

Minor points:

- Figure 1C: it would be more convincing if the authors could show the interaction also with endogenous proteins, not only upon overexpression of FANCI.

Co-IP of the endogenous FANCI and AND-1 proteins has been carried out and the results are shown in Figure 1C.

- Figure 2B: Why is there no FANCI-WT protein present in lane 3 of the western blot?

In lane 3 of Figure 2B (in this version it corresponds to Figure 2C) FANCI is not present as this lane contains the pulled-down sample of a negative control experiment, where the AND-1 6xHis SepB domain was incubated with the anti-Flag beads in the absence of the FANCI protein.

- Figure 2: Considering the fact that there are several CIP-box proteins that interact with AND1 it would be interesting to test an additional one in the in vitro binding assays. Doing competition assays they could say something about the relative binding strength.

This experiment has not been carried out.

- Figure 3: This figure could go to the supplement. It shows the generation of reagents and repeats data that were already known.

In the present manuscript data related to the generation of the FANCF-KO cell line using the CRISPR-Cas9 methodology are in Figure EV3.

- Figure 5B: why is there so much Edu-FANCF PLA signal outside the nucleus? Based on that, how can we be sure the signal within the nucleus is specific? Also, what is the function of the Edu-Edu PLA control? A better positive control would be to test PLA of Edu with a known replisome component.

The EdU-FANCF PLA spots detected outside the nuclei in all the conditions tested in Figure 5B (images on the bottom) are likely due to the not optimal quality of the commercial anti-FANCF antibody. In fact, these background spots are not observed in the EdU-EdU PLA experiment (upper images of the same Figure Panel). However, the area of the nuclei has been selected for spot quantification. Moreover, as pointed out in the text at page 12, EdU-EdU proximity ligation assays are carried out to verify if all the tested cell lines have a similar number of active replication factories and equal probability to produce a positive signal in the conditions used for the SiRF assays, as suggested in the paper, where this technique was first described (Roy S, Luzwick JW & Schlacher K (2018) SiRF: Quantitative in situ analysis of protein interactions at DNA replication forks. *J Cell Biol* 217: 1553–1553).

- Figure 6A: The numbers 'floating' under the graph are not easy to interpret. It would be better to add a table with these numbers for 'median fork speed' and 'number of tracks measured'.

We do not believe adding a Table with the numbers reported in the indicated Figure (now Figure 5A) is necessary.

- Figure 7B: In the legend it is stated: 'Level of immunoprecipitated AND-1 was normalized to pulled-down Flag-tagged FANCF in each sample'. Based on the quantification of the results of the Q740H mutant in the western blot I think this is not what was done. The FANCF signal in the IP for this mutant seems pretty much equal in intensity compared to the AND-1 IP signal. This ratio seems to be very similar to the WT FANCF condition (in fact there seems to be less AND-1 signal in wt FANCF). Therefore, I suspect that the 'input' signal of FANCF is used for the quantification. This is very relevant there seems to be no difference in AND-1 interaction with the Q740H mutant if you quantify the way described in the figure legend.

In response to this point, we have tried to improve quality of the co-pull-down experiments shown in Figure 7B. It should be considered that expression level of FANCF-Flag proteins varies among different mutant derivatives and for each mutant among different transient transfection experiments, making it difficult to have Western blots, where the amounts of the ectopically expressed FANCF proteins are equal in all the input and pulled-down samples. However, our quantitative analysis, based on at least three biological replicates of the experiment, has revealed that association of FANCF AALA, N734H and D736H mutants to endogenous AND-1 is severely reduced as compared to the FANCF WT protein.

Referee #3:

The manuscript by Boavida et al. shows that the DNA helicase FANCF interacts with AND-1, a protein adaptor that localizes at the replication fork. The interaction is facilitated via a CIP box in FANCF and the SepB domain in AND-1. Mutating the CIP box of FANCF impairs the binding between the two proteins but does not alter FANCF's helicase activity. This is also the case for patient-derived mutations in the CIP box. The authors suggest that this interaction allows FANCF to be recruited to replication forks.

While the interaction between FANCF and AND-1 is noteworthy, I don't find this manuscript a particularly strong candidate for EMBO J. The advance is incremental in nature and the data does not thoroughly show that the interaction between FANCF and AND-1 is relevant for its function at the replication fork. There are also a few problems with the interpretation of the data, which does not seem appropriate based on the data presented.

We would like to thank Referee #3 for his helpful comments that we have addressed as described below.

Major points:

Figure 3: FJ-KO complemented with WT or mutants should be included in the viability assays to show functional relevance of FANCF/AND-1 interaction

This point has been already addressed, as above described (see reply to point 2 of Referee # 2).

No data are provided to demonstrate that the interaction between FANCF and AND-1 is relevant for its function at the replication fork. The authors should also try PLA in AND-1 mutant cells and assess its impact on FANCF binding to the fork.

In our opinion, DNA fiber track analysis reveal the relevance of the FANCI/AND-1 interaction for the FANCI functions at the replication fork. These experiments, reported in Figure 5, indicate that in the absence of FANCI or when FANCI is not associated to the replisome (due to the CIP box AALA mutation), replication forks are slowed down, fork recovery is reduced, fork asymmetry is increased, either in unchallenged conditions or after treatment with Pyridostatin.

We believe that experiments in cell lines where AND-1 is mutated to abolish the interaction with FANCI would be difficult to carry out and to interpret, since AND-1 interacts not only with FANCI but also with key replication factors in the core replisome (such as DNA polymerase α and the CDC45-MCM2-7-GINS complex, both bound to AND-1 *via* CIP-box-mediated interactions). Therefore, altering the AND-1 protein interaction hub is expected to compromise the assembly and stability of the whole replication machinery with detrimental effects for cell proliferation and viability. Therefore, while we believe these experiments would not add any substantial novel information to our work, it might also lead to misinterpretations.

FANCI is also required for fork protection in replication stress condition (Peng et al. Cell report_2018). Is the interaction between AND-1 and FANCI required for fork protection?

We agree with Referee # 3, this would be an important issue to clarify and could be addressed in a subsequent work.

Many western blots are suboptimal: missing loading ctrl/input, uneven loading, which makes it difficult to compare samples

As previously described (see reply to the last minor point raised by Referee # 2), quality of Western blot experiments has been improved in this version of the manuscript.

Gel in 7D: loading is not consistent. Also, the results are described as comparable between WT and mutant proteins but the kinetics differ significantly.

As previously described (see reply to Referee # 2), the quality of the DNA helicase assays has been improved and additional controls have been added in the present version of the manuscript.

Minor comments:

Increase the font size in all the figures

We have made this modification in the current version of the manuscript

The start of the results section it is sometimes difficult to follow with the discussion of Chl1 and Ddx11. This could be better written/explained.

We have elaborated on this topic on the *Discussion* section of the present manuscript.

The structure of the paper is somewhat confusing with the authors referring to Fig. 7 in paragraph 1 and referring to similar sequence alignments at different times.

We have decided to show the DNA helicase assays for all the FANCI proteins described in this work (FANCI WT, AALA, K52R, N734H and D736H) in the same Figure to make easier a direct comparison among them. This is the reason why we refer to Figure 7 (the last main Figure of the manuscript) in the first paragraph of the *Results* section.

In addition, to better substantiate our key finding that FANCI directly binds AND-1 and this interaction is mediated by the newly-identified FANCI CIP box, we have expressed and purified full-length AND-1 from insect cells and we have demonstrated that full-length AND-1 directly binds to FANCI wild type but not the AALA mutant derivative. The results have been shown in Figure 2B and EV2A.

I would like to point out that most of the issues raised by the three referees have been addressed by us and we are ready to submit a revised version of the manuscript that includes all the additional data in Expanded View Figures, as described in the above point-by-point reply.

Dear Francesca,

Thank you for the transfer of your revised manuscript to EMBO reports. We have now heard back from referee 2, who supports the publication of the revised study if her/his last concerns can be addressed. Please find the report below.

I would thus like to invite you to address the last few referee comments. In addition, a few editorial requests will also need to be addressed:

- Please submit with your revised ms a completed author checklist that can be found here:

<<https://www.embopress.org/page/journal/14693178/authorguide>> The completed checklist will also be part of our transparent peer-review process file (RPF).

- All main and all EV figures need to be uploaded as individual, high resolution figure files. Expanded View Figure 1, etc should be renamed to Figure EV1, etc.

- Please add up to 5 keywords to your ms file.

- Please add a DATA AVAILABILITY SECTION (DAS) to the end of the materials and methods. If you have not deposited any newly generated data in public databases, please mention this fact in the DAS.

- Please add all funding info also in our online submission system. Some info is currently missing.

- Please remove the author credits from the ms file. All credits need to be entered online during ms submission.

- Please remove "data not shown" on page 46 as per journal policy.

- FIGURE CALLOUTS are missing for Fig 6A,B ; Fig 7A is called out before Fig 2 and before Fig 6; please correct.

- Please add the legend for Fig 7.

- My colleague Hannah will contact you with requests for source data that need to be provided with the re-submission of your ms.

- Please address the following comments from our data editors:

1. Please note that the measure of center for the error bars needs to be defined in the legend of figure 1d.
2. Please note that in figure 3 the scale bar unit should be corrected from μM to μm .
3. Please note that the scale bar needs to be defined for figure 4b
4. Please indicate what arrowhead represents in the legend of figure 5a

EMBO press papers are accompanied online by A) a short (1-2 sentences) summary of the findings and their significance, B) 2-3 bullet points highlighting key results and C) a synopsis image that is exactly 550 pixels wide and 200-600 pixels high (the height is variable). You can either show a model or key data in the synopsis image. Please note that text needs to be readable at the final size. Please send us this information along with the final manuscript.

Kind regards,
Esther

Referee 2:

The authors performed new helicase assays and included a dominant negative mutant for FANCI. This has much improved and I am satisfied with this.

They performed gamma-H2AX foci formation assays on some of the cancer associated mutants and showed that these are enhanced. This further supports that disrupting this interaction could be linked to cancer risk. This data should be added to the

manuscript.

Experiments are not performed in another cell type but I can be fine with this since I agree that this would take a lot of time. They performed FANCD1 (WT and mutants) IP's and tested binding to other DNA repair proteins: BRCA1, BLM and MLH1. Although there does seem to be slight differences in binding efficiency I do agree with the authors that all these proteins are probably still capable of interacting with FANCD1. Although maybe not completely solid this can provide an indication that other pathways are not affected. This data should be added to the manuscript.

To me, 2 major points are still not fully addressed:

As you also mention, they did not do proper survival assays for the interaction mutant(s). In the rebuttal they show proliferation assays and although they are interesting, these are difficult to interpret. Basically, they show that all cell lines tested are sensitive to MMC and Pyridostatin but the relative differences between cell lines is not so clear. To my opinion this should really be tested in a survival assay.

The authors have not shown conclusively that the interaction between AND-1 and FANCD1 is constitutive. This could still be induced by (endogenous) damage/replication fork barriers. However, if they clearly state this option in the text and remove the conclusion that this interaction is constitutive I am fine with this.

On last point that has not been properly addressed is the quantification of the western blot in Figure 7 (as was also raised by reviewer 3). The authors have now replaced this blot with a different experiment that shows quite different results from the previous version. Although I understand that obtaining similar expression levels is challenging and therefore there might be a large error bar on these experiments, it should at least be clearly explained how this quantification is performed. Currently, it is stated in the figure legend that the 'Level of immunoprecipitated endogenous AND-1 was normalized to pulled down Flag-tagged FANCD1 WT in each sample'. This is confusing as there is no FANCD1 WT in each sample. The correct way to normalize these blots to my opinion is to normalize to the pulled down Flag-tagged FANCD1 (WT or mutant, in the same lane, depending on the condition). According to the quantification I have a hard time believing that that is done.

Altogether, I think this study is interesting and I would recommend acceptance but only when a proper survival assay is performed and the issues with the western blot are clarified.

Dear Esther,

I would like to thank you for your help and assistance with the revision of our manuscript.

A point-by-point response to all the issues raised by the Referee can be found below.

Referee 2:

The authors performed new helicase assays and included a dominant negative mutant for FANCD1. This has much improved and I am satisfied with this.

We are happy the Referee has appreciated the efforts we made to produce additional FANCD1 mutants and biochemically characterize them in the revised version of the manuscript.

They performed gamma-H2AX foci formation assays on some of the cancer associated mutants and showed that these are enhanced. This further supports that disrupting this interaction could be linked to cancer risk. This data should be added to the manuscript.

We have added the data on the gamma-H2AX focus formation for the cancer-related FANCD1 CIP box variants (FANCD1 N734H and D736H) in the Figure EV4B.

Experiments are not performed in another cell type but I can be fine with this since I agree that this would take a lot of time.

They performed FANCD1 (WT and mutants) IP's and tested binding to other DNA repair proteins: BRCA1, BLM and MLH1. Although there does seem to be slight differences in binding efficiency I do agree with the authors that all these proteins are probably still capable of interacting with FANCD1. Although maybe not completely solid this can provide an indication that other pathways are not affected. This data should be added to the manuscript.

The results of the co-IP experiments of FANCD1 WT and CIP box mutants with BRCA1, BLM and MLH1 are shown in Figure EV4A.

To me, 2 major points are still not fully addressed:

As you also mention, they did not do proper survival assays for the interaction mutant(s). In the rebuttal they show proliferation assays and although they are interesting, these are difficult to interpret. Basically, they show that all cell lines tested are sensitive to MMC and Pyridostatin but the relative differences between cell lines is not so clear. To my opinion this should really be tested in a survival assay.

We agree with Referee 2 that the proliferation assays do not clearly reveal a difference in drug sensitivity among the various cell lines. As suggested, we have carried out survival assays and the results are now shown in Figure EV3B. The data reveal that re-expressing either FANCD1 WT or the AALA mutant in FJ-KO cells reverts sensitivity to both drugs (even if the rescue by the FANCD1 AALA mutant is not complete after PDS treatment at the higher concentrations used). These results indicate that alternative DNA repair pathways are activated to counteract enhanced DNA damage caused by faulty association of FANCD1 to DNA replication forks.

The authors have not shown conclusively that the interaction between AND-1 and FANCD1 is constitutive. This could still be induced by (endogenous) damage/replication fork barriers. However, if they clearly state this option in the text and remove the conclusion that this interaction is constitutive I am fine with this.

We agree with Referee 2 that we cannot rule out the possibility that endogenous DNA damage or replication fork obstacles could trigger recruitment of FANCD1 to the replisome by AND-1. This is now clearly stated in the text (last paragraph at page 13).

On last point that has not been properly addressed is the quantification of the western blot in Figure 7 (as was also raised by reviewer 3). The authors have now replaced this blot with a different experiment that shows quite different results from the previous version. Although I understand that

obtaining similar expression levels is challenging and therefore there might be a large error bar on these experiments, it should at least be clearly explained how this quantification is performed. Currently, it is stated in the figure legend that the 'Level of immunoprecipitated endogenous AND-1 was normalized to pulled down Flag-tagged FANCI WT in each sample'. This is confusing as there is no FANCI WT in each sample. The correct way to normalize these blots to my opinion is to normalize to the pulled down Flag-tagged FANCI (WT or mutant, in the same lane, depending on the condition). According to the quantification I have a hard time believing that that is done.

We thank the Referee for pointing out this issue and we admit that Western blot bands quantification was not described in a clear and detailed manner leading to misinterpretation of the data. We have now added a detailed explanation of the method used to perform the band quantification in the *Materials and Methods* section (at page 25-26).

Altogether, I think this study is interesting and I would recommend acceptance but only when a proper survival assay is performed and the issues with the western blot are clarified.

All the other editorial requests have been also addressed in the revised version.

We hope our manuscript deserves publication in EMBO reports.

Regards

Francesca Pisani

Dr. Francesca Pisani
Istituto di Biochimica e Biologia Cellulare - CNR
Via P. Castellino, 111
Napoli, NA 80131
Italy

Dear Francesca,

I am very pleased to accept your manuscript for publication in the next available issue of EMBO reports. Thank you for your contribution to our journal.
